# Axl and MerTK regulate synovial inflammation and are modulated by IL-6 inhibition in rheumatoid arthritis

Alessandra Nerviani [1,6], Marie-Astrid Boutet[1,2,6], Giulia Maria Ghirardi [1,6], Katriona Goldmann[1], Elisabetta Sciacca [1], Felice Rivellese [1], Elena Pontarini [1], Edoardo Prediletto [1], Federico Abatecola [1], Mattia Caliste[1], Sara Pagani[1], Daniele Mauro[1], Mattia Bellan[1,3], Cankut Cubuk[1], Rachel Lau [1], Sarah E. Church [4], Briana M. Hudson[4], Frances Humby[1], Michele Bombardieri[1], Myles J. Lewis [1] & Costantino Pitzalis [1,5] ✉

The TAM tyrosine kinases, Axl and MerTK, play an important role in rheumatoid arthritis (RA). Here, using a unique synovial tissue bioresource of patients with RA matched for disease stage and treatment exposure, we assessed how Axl and MerTK relate to synovial histopathology and disease activity, and their topographical expression and longitudinal modulation by targeted treatments. We show that in treatment-naive patients, high *AXL* levels are associated with pauci-immune histology and low disease activity and inversely correlate with the expression levels of pro-inflammatory genes. We define the location of Axl/MerTK in rheumatoid synovium using immunohistochemistry/ fluorescence and digital spatial profiling and show that Axl is preferentially expressed in the lining layer. Moreover, its ectodomain, released in the synovial fluid, is associated with synovial histopathology. We also show that Toll-like-receptor 4-stimulated synovial fibroblasts from patients with RA modulate MerTK shedding by macrophages. Lastly, Axl/MerTK synovial expression is influenced by disease stage and therapeutic intervention, notably by IL-6 inhibition. These findings suggest that Axl/MerTK are a dynamic axis modulated by synovial cellular features, disease stage and treatment.

Tyro3, Axl, and MerTK are the three members of the TAM tyrosine kinase receptor family. Historically, the deletion of all three TAM receptors in triple knock-out mice caused the development of broad-spectrum autoimmunity, including destructive inflammatory arthritis of the paws[1]. Since then, Axl and MerTK, in particular, have been implicated in the pathogenesis of human autoimmune diseases, including rheumatoid arthritis (RA); however, the exact contribution of individual TAMs toward the regulation of inflammation and clinical response to therapy remains to be elucidated.

Data from animal models suggest that Axl would play a physiological protective role in the joint[2]. Indeed, elegant fate-mapping experiments have confirmed that Axl is expressed by a subset of synovial lining CX3CR1[+] macrophages, which behaves as a physical barrier that secludes the articular cavity and becomes disrupted when

[1]Centre for Experimental Medicine and Rheumatology, William Harvey Research Institute, Barts and The London School of Medicine and Dentistry, Queen Mary University of London & NIHR BRC Barts Health NHS Trust, London, UK. [2]Nantes Université, Oniris, INSERM, Regenerative Medicine and Skeleton, RMeS, UMR 1229, F-44000 Nantes, France. [3]Department of Rheumatology, University of Eastern Piedmont and Maggiore della Carita Hospital, Novara, Italy. [4]NanoString Technologies Inc, Seattle, WA, USA. [5]Department of Biomedical Sciences, Humanitas University & IRCCS Humanitas Research Hospital, Milan, Italy. [6]These authors contributed equally: Alessandra Nerviani, Marie-Astrid Boutet, Giulia Maria Ghirardi. ✉e-mail: c.pitzalis@qmul.ac.uk

an inflammatory pro-arthritic process is initiated[3]. Interestingly, peripheral blood and synovial fluid CD1c+ dendritic cells (DCs) from individuals with RA showed a constitutive overexpression of miRNA-34a, responsible for inhibiting Axl expression, and were less capable of limiting inflammatory responses compared to healthy donors[4]. While early work in human synovial tissue found that Axl was present in the lining, around blood vessels, and could be detected in synovial fluid (SF)[5], more recently, soluble Axl has been confirmed to be one of the most abundant proteins in RA SF versus non-RA[6].

A protective role for MerTK has also been described in experimental models of arthritis and confirmed in micromasses of human monocytes and synovial fibroblasts. In fact, blocking MerTK drove increased pro-inflammatory cytokine release, worsening of pathology and more severe joint symptoms secondary to the inhibition of MerTK-mediated efferocytosis[7]. Consistently, MerTK is preferentially expressed by alternatively activated anti-inflammatory/regulatory macrophages (M2)[8]. Single-cell (sc) RNA-sequencing (RNA-seq) analysis of synovial tissue suggested that high expression of MerTK characterised non-inflammatory arthritis macrophages (e.g., osteoarthritis)[9] and, when co-expressed with CD206, MerTK identified a subset of synovial macrophages more abundant in healthy joints and RA patients in remission[10].

Despite sharing significant structural homology and having ligands in common, in vitro experiments showed that Axl and MerTK expression and functions were remarkably diverse in bone marrow-derived murine DCs and macrophages; while MerTK expression was enhanced in homoeostatic/anti-inflammatory conditions following dexamethasone induction, Axl, on the contrary, was upregulated in response to pro-inflammatory stimuli[11].

A growing body of evidence suggests that Axl and MerTK play a crucial role in RA pathogenesis and progression and may be exploited as novel therapeutic targets. This is fundamentally important because, notwithstanding the significant improvement in the management and prognosis of RA, a substantial 30−40% of patients still do not respond adequately to treatment in a timely manner.

In this study, we aim to address a number of critical unanswered questions taking advantage of a unique synovial tissue biomedical resource of well-characterised RA patients matched for disease stage and treatment exposure, emerging from the Pathobiology of Early Arthritis Cohort (PEAC) and the first worldwide biopsy-driven randomised clinical trial: Rituximab versus tocilizumab in anti-TNF inadequate responder (ir) patients with rheumatoid arthritis (R4RA).

We provide insights into Axl and MerTK relationship to synovial pathotypes and disease activity, their detailed topographical expression, how stromal and myeloid cells interact to regulate Axl and MerTK expression and, most importantly, their modulation longitudinally by treatment intervention with the IL-6 receptor inhibitor tocilizumab.

## Results

### Higher synovial *AXL* transcript levels are associated with pauci-immune histology, lower disease activity, and inversely correlate with pro-inflammatory genes

To assess the relationship between *AXL*/*MERTK* gene expression, synovial pathotypes and disease activity, we first quantified synovial *AXL* and *MERTK* transcripts in a cohort of 87 early treatment-naive patients[12]. The baseline demographics, clinical and histology features are described in Supplementary Table 1. Notably, at disease presentation, in the absence of previous treatment exposure, *AXL* was significantly upregulated in pauci-immune patients, defined by the lack of immune cell infiltration, compared with diffuse- and lympho-myeloid patients, characterised by abundant inflammation. *MERTK*, on the other hand, did not show a preferential expression in any of the three pathotypes (Fig. 1A). Consistently, *AXL*, but not *MERTK*, was negatively correlated with all markers of immune cell infiltration within the tissue (Fig. 1B). There was no significant correlation between *AXL*

and *MERTK* (not shown, $r = 0.1$, $p$ 0.15), including when this was assessed in relation to the clinical response to conventional synthetic (cs) Disease Modifying Anti-Rheumatic Drugs (DMARDs) treatment at six months (Fig. 1C). Further analysis based on synovium modules defined by weighted gene correlation network analysis (WGCNA), as described in[12], showed that while *AXL* is included in the metabolic pathways module (sc151), characterising pauci-immune and myeloid synovial tissues, *MERTK* is encompassed in the macrophage module (sc209), which is upregulated in lympho- and diffuse-myeloid patients (Fig. 1D). Importantly, synovial *AXL* (but not *MERTK*) inversely correlated with several pro-inflammatory genes like tumor necrosis factor (*TNF*), Interleukin (*IL*)−6, *IL-1B* and C-C motif Chemokine Ligand 8 (*CCL8*) but positively with its known enhancer Transforming Growth Factor beta (*TGFb*) (Fig. 1E). As the exposure to different growth factors prompts monocytes to differentiate towards alternative activation states and Axl and MerTK expression is influenced by macrophages/DCs phenotype[11], we next analysed the relationship between synovial *AXL*/*MERTK* and several critical monocyte/macrophage-growth factors. We observed that each receptor has a distinctive correlation pattern with these key molecules, e.g., *AXL* positively correlates with Colony Stimulating Factor 1 (*CSF1*), encoding for M-CSF, which drives alternatively activated macrophages, but negatively with *CSF2*, encoding for GM-CSF, promoting inflammatory polarisation, while *MERTK* shows a positive correlation with both *CSF1* and *CSF2* receptors (Fig. 1E). In order to establish the clinical significance of *AXL* and *MERTK* synovial gene expression, we then investigated their relationship with disease activity. Notably, a lower level of Axl synovial gene expression correlated with a higher 28 joint count disease activity score (DAS28) and markers of systemic inflammation such as erythrocyte sedimentation rate (ESR) and C-reactive protein (CRP) (Fig. 1F).

### *AXL* and *MERTK* STRING-defined gene-network modules cluster with synovial histopathology and disease activity

To further investigate the inter-relationship of *AXL* with *MERTK* as well as the up- and downstream genes linked to either, we defined Axl and MerTK-specific modules composed of 31 predicted partners using STRING network analysis (Supplementary Fig. 1A and Supplementary 1B). We found an expected substantial overlap between these sets: 13 genes were common to both modules and included Axl/MerTK ligand Growth Arrest-Specific gene 6 (*GAS6*), MerTK ligand Protein S (*PROS1*), and the Epidermal-Growth-Factor-Receptor (*EGFR*). Conversely, 18 genes were uniquely present in the Axl or the MerTK module. The former was characterised by *PIK3CA*/*PIK3CB*/*PIK3R1*, encoding for the Phosphoinositide-3-Kinase (PIK3) catalytic subunits p110α/p110β and the regulatory subunit p85α, respectively; *IGF1R*, encoding for the Insulin-Growth-Factor-Receptor1; *IFNAR1*, encoding for Interferon α-and-β-Receptor-Subunit1, and the Signal Transducer and Activator of Transcription (*STAT*)−3. MerTK module included the recently discovered MerTK-ligands *TULP* and *LGALS3*, encoding for Galectin 3, a crucial molecule in the synovial microenvironment interaction[13]; *CD64*, encoding for Fc-gamma receptor 1A (FcγR1A); and *CD28*, encoding for the CD80/CD86 receptor (Supplementary Fig. 1C and Supplementary Fig. 1D). Several but not all genes included in the Axl module showed various degrees of correlation with the genes defining the MerTK module and vice-versa, suggesting the existence of both common as well as receptor-specific pathways (Supplementary Fig. 1E).

We then quantified the expression of each module in the synovial tissue from early arthritis treatment-naive patients ($n = 81$ patients with both RNA-seq and histological classification available). As shown in Fig. 2A, based on the relative expression of the Axl module, we identified three clusters of patients: i. one characterised by lympho-myeloid pathotype, higher synovitis scores, more patients with active disease (DAS28 > 5.1), and defined by the upregulation of Axl gene-

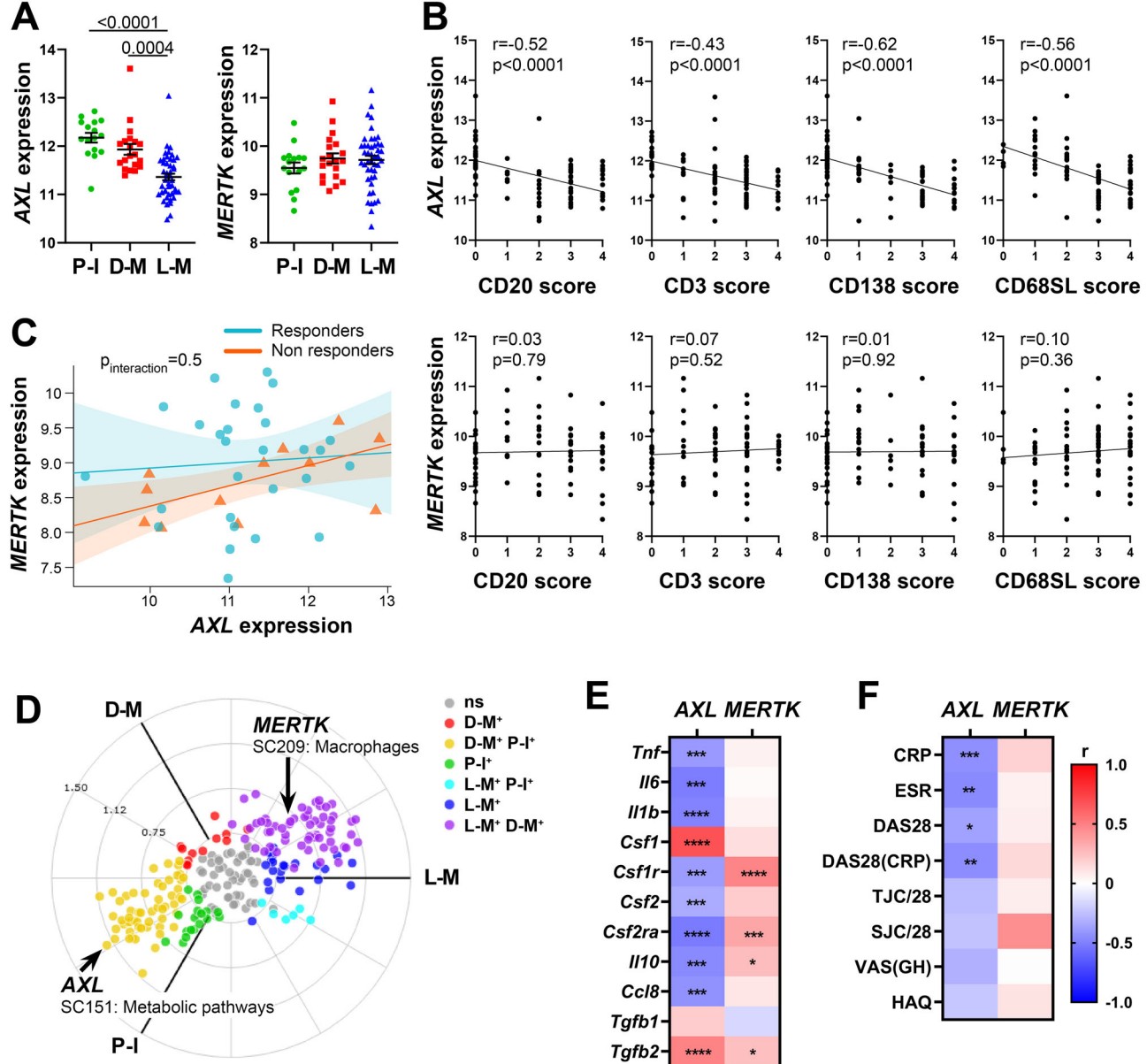

**Fig. 1 | Synovial *AXL* and *MERTK* have distinct molecular expression patterns in early untreated RA patients. A** Axl and MerTK gene expression (regularised-log normalised reads) in synovial tissue of early arthritis treatment-naive RA patients ($n = 87$) according to the histological pathotype defined as Pauci-Immune (P-I, in green), Diffuse-Myeloid (D-M, in red), and Lympho-Myeloid (L-M, in blue). Data are represented as mean ±SEM. *p* values indicated were calculated using the Kruskal–Wallis test with Dunn's post hoc test. **B** Correlation between synovial *AXL* (top panels) and *MERTK* (bottom panels) gene expression and semi-quantitative scores (0–4) of B cells (CD20), T cells (CD3), plasma cells (CD138), and sublining macrophages (CD68SL). *p* values and *r*-coefficients were calculated using the two-tailed Pearson correlation test. **C** Regression model analysis with interaction term to estimate the correlation of *AXL* with *MERTK* expression in relation to clinical response to conventional synthetic Disease-Modifying-Anti-Rheumatic-Drugs. The clinical response was assessed by EULAR criteria with DAS28(CRP) after 6-months of treatment (good responders in light blue; moderate and non-responders in orange). p-interaction is not significant. The scatter plots show the regression line

of the fitted negative binomial generalised mixed effects model with the error bars showing 95% confidence interval (fixed effects). **D** 2D polar plot of transcript modules containing *AXL* and *MERTK* in synovial tissue characterised by lympho-myeloid (L-M), diffuse-myeloid (D-M), and pauci-immune (P-I) pathotypes. Different colours show pairwise comparisons between the three pathotypes: upregulation in one group only (D-M: red, P-I: green and L-M: blue) or in two groups (D-M/P-I: yellow, L-M/P-I: light blue, L-M/D-M: purple). **E, F** Heatmaps showing the correlation between *AXL* and *MERTK* synovial transcript levels at baseline and cytokines and growth factors relevant to the inflammatory response (**E**) and clinical parameters (**F**). The red/blue scale represents the Spearman *r* coefficient, calculated using the two-tailed Spearman correlation test. \**p* < 0.05, \*\**p* < 0.01, \*\*\**p* < 0.001, \*\*\*\**p* < 0.0001. CRP, C-Reactive Protein; ESR, erythrocyte sedimentation rate; DAS28 disease activity score 28; TJC/28, tender joints count (0-28); SJC/28, swollen joints count (0-28); VAS GH, Visual Analogue Scale General Health (0–100); HAQ health assessment questionnaire.

partners involved in T-cell activation (*ZAP70, LCP2*), B-cell development (*BLNK*) and interferons alpha and beta signalling (*IFNAR1*); ii. one including patients with pauci-immune and diffuse-myeloid pathotypes, lower synovitis scores, predominantly low- and moderate-disease activity (DAS28 < 5.1), defined by the upregulation of *AXL*,

*PROS1, IGF1R, CDH11, EGFR,* and *ERBB* genes; iii. and the third a mixed cluster characterised by both lympho- and diffuse-myeloid pathotypes, with intermediate synovitis scores and disease activity status (Fig. 2A). Considering the amount of gene overlap between the two modules, expectedly, the expression of the MerTK module (Fig. 2C)

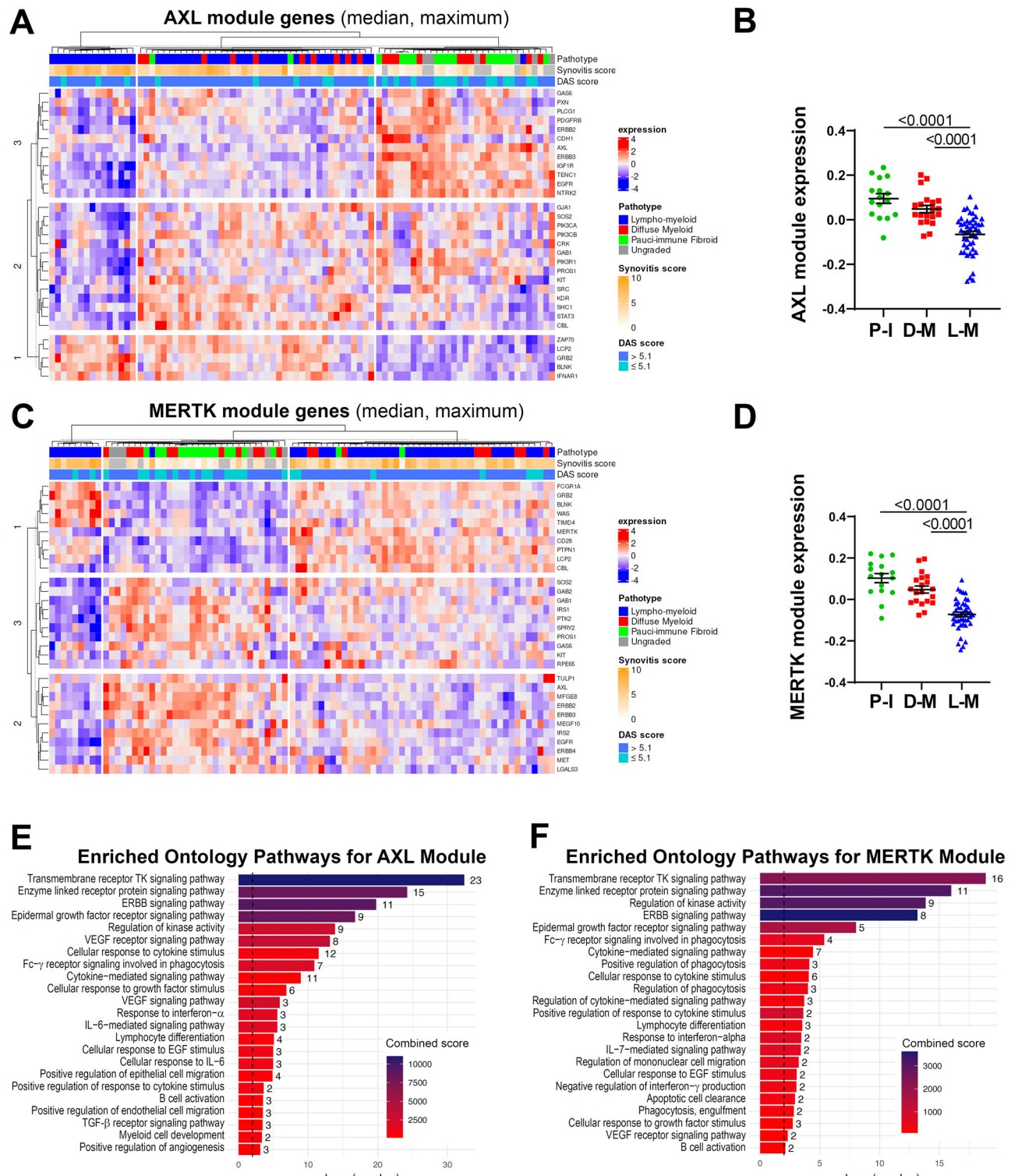

**Fig. 2 | *AXL* and *MERTK* STRING networks similarly correlate with synovial histology. A, C** Heatmaps showing regularised-log-transformed expression for all genes in the Axl module (**A**) and the MerTK module (**C**), (**B, D**) Axl module (**B**) and MerTK module (**D**) expression in synovial tissue of early arthritis treatment-naive RA patients (*n* = 81) according to the histological pathotype defined as Pauci-Immune (P-I, in green), Diffuse-Myeloid (D-M, in red), and Lympho-Myeloid (L-M, in blue). Data are represented as mean ±SEM. *p* values indicated were calculated using the Kruskal–Wallis test with Dunn's post hoc test. **E, F** Enriched ontology pathways for the Axl module (**E**) and MerTK module (**F**). Nominal *p* values from gene set enrichment analysis are shown.

also clustered in three groups: i.e., one characterised by predominant lympho-myeloid pathotype, higher synovitis scores, DAS28 values predominantly >5.1, and upregulation of MerTK gene-partners encoding for proteins key in survival, proliferation, and activation of T cells (*CD28, LCP2*), B-cell activation (*BLNK*), and immune responses (*PTPN1, FCGR1A*); ii. one mainly containing pauci-immune and diffuse-myeloid pathotypes, lower synovitis scores, low- and moderate-disease activity (DAS28 < 5.1), and characterised by the upregulation

of *AXL*, TAM receptors' ligands *LGALS3* and *PROS1*, *EGFR* and *ERBB* genes; iii. and the third cluster without clear pathotypes, synovitis scores and disease activity scores signature. The overall expression of both Axl and MerTK modules was significantly higher in the pauci-immune cell-poor synovial tissue compared with the cell-rich lympho- and diffuse-myeloid (Fig. 2B, D), indicating that quantitative gene expression is not simply the result of the level of synovial cellular infiltration.

Using gene set enrichment analysis (GSEA), we observed a substantial overlap of pathways enriched in both Axl and MerTK modules, including Tyrosine Kinases, ERBB, FcγR signalling pathways, and response to Interferon-α (Fig. 2E, F). Notably, however, the Axl module was distinctively linked to the IL-6-mediated signalling pathway and cellular response to IL-6, TGF-β receptor signalling, and regulation of angiogenesis (Fig. 2E). Conversely, the MerTK module was associated specifically with enrichment in phagocytosis and apoptotic cells clearance pathways (Fig. 2F).

Taken together, these data showed that Axl/MerTK gene expression, as well as Axl-/MerTK STRING modules and related pathways, are linked to synovial histopathology.

### Axl is preferentially expressed by lining layer cells and its ecto-domain can be cleaved and released in the synovial fluid

As topographical distribution is often associated with positional function, we investigated Axl and MerTK expression at the protein level within the distinct lining and sublining areas of the RA synovial tissue. We show that, in early treatment-naive RA synovia, Axl was predominantly expressed within the lining layer and barely found in the sublining region (Fig. 3A), in line with data in animal models and patients with established late RA[2,3]. Since the lining is composed of macrophages and fibroblasts-like-synoviocytes (FLS), we performed multiplex-Immuno Fluorescence (IF) staining with the pan-macrophage marker CD68 and the fibroblast markers CD55 (typically expressed by lining-FLS) to confirm lineage-specific Axl surface expression. As shown in Fig. 3B, Axl is expressed by either CD68[+] lining macrophages or CD55[+] lining FLS. As fibroblasts are known to have a positional identity within the tissue, represented by distinct surface markers[9,13], we next confirmed that Axl expression was specific to CD55[+] lining fibroblasts, while not detected in sublining CD90/Thy[+] stromal cells (Supplementary Fig. 2A). Recent work within the accelerating medicines partnership (AMP) consortium showed that at least four FLS subsets, characterised by specific markers, could be identified in RA synovium by scRNA-seq, in line with our protein data, we confirmed that *AXL* was higher in the CD55 + SC-F4 (Supplementary Fig. 2B), more abundant in the lining and leucocyte-poor RA (Supplementary Fig. 2C). Also importantly, despite the marked infiltration of CD68[+] cells in the sublining, most interstitial macrophages did not express Axl on their surface (Fig. 3B).

Conversely, MerTK was predominantly expressed by CD68[+] macrophages. Like Axl, MerTK was preferentially found in the lining but also expressed in the sublining (Fig. 3C) and in the lymphocytic aggregates by tingible-body-macrophage-like cells. Macrophages of the lining could express either MerTK alone, Axl alone or co-express MerTK and Axl, suggesting that distinct macrophage lineage states co-exist in the lining (Fig. 3D).

Given the ability of Axl and MerTK ectodomains to be cleaved by metalloproteases[11], and in keeping with previous reports showing that soluble Axl (sAxl) is abundant in the synovial fluid of RA patients[6], we confirmed that both Axl-bearing and Axl-negative cells expressed the cleaving enzyme ADAM10, including within the synovial region facing the joint cavity (Fig. 3E). Next, we quantified the soluble extracellular domain of Axl (sAxl), MerTK (sMerTK), and their ligand Gas6 in the SF obtained at the same time as the synovial biopsy in a subset of early arthritis untreated patients (Supplementary Table 2). In these matched samples, we demonstrated that SF sAxl was present in excess

compared to sMerTK and its ligand Gas6 (Fig. 3F), in line with Axl's role as a decoy receptor[14]. Interestingly, SF sAxl levels, but not sMerTK or Gas6 levels (Supplementary Fig. 3A, B), were significantly raised in patients with a high degree of synovial inflammation (Fig. 3G), suggestive of an unsuccessful homoeostatic reaction attempting to restrain tissue inflammation. Only SF sAxl also positively correlated with markers of systemic inflammation like ESR (Fig. 3H and Supplementary Fig. 3B, D), further corroborating that Axl cleavage may play an immuno-modulatory role as previously proposed in other auto-immune diseases.

### Axl and MerTK synovial expression in relationship with clinical features are influenced by the disease stage and treatment exposure

To define whether the synovial expression of *AXL* and *MERTK* varies according to the disease stage and is modulated by treatment exposure, we compared *AXL* and *MERTK* RNA expression from the early arthritis cohort to a cohort of late, difficult-to-treat (D2T) RA patients, characterised by an inadequate response (ir) to csDMARDs and at least one TNFα inhibitor (TNFα-ir) prior to randomisation to a second-line biologic agent (either rituximab -RTX- or tocilizumab -TOC) in the R4RA biopsy-driven randomised clinical trial, described in refs. 15,16; data available at https://r4ra.hpc.qmul.ac.uk/.

At this late D2T disease stage, similarly to the early phase (Fig. 1A, B), *AXL* was significantly higher in low-inflamed tissues (pauci-immune) (Fig. 4A) and significantly negatively correlated with all markers of immune cell tissue infiltration (Fig. 4B). *MERTK* was instead significantly upregulated in diffuse-myeloid and lympho-myeloid patients (Fig. 4C) and positively correlated with the inflammatory cellular infiltrate (Fig. 4D), suggesting that the exposure to medications such as corticosteroids, csDMARDs, and TNF inhibitors and/or uncontrolled tissue inflammation might alter MerTK expression. To further elucidate how different cell types (stromal, myeloid) influence the modulation of Axl and MerTK receptors in an inflamed environment, we set up two in vitro systems of macrophage-FLS interaction using conditioning media, as described in Supplementary Fig. 4A. We observed that macrophages did not influence sAxl, sMerTK, or Gas6 release by conditioned RA-FLS, independently of the macrophage polarisation status (Fig. 4E and Supplementary Fig. 4B); conversely, macrophages conditioned with TLR4-stimulated RA-FLS significantly decreased sMerTK release in the supernatants. Consistently, sMerTK/sAxl ratio was significantly lower when compared with conditioning by unstimulated FLS (Fig. 4F).

At this late stage of RA, similarly to what we observed in the early treatment-naive cohort (Fig. 1E), *AXL* maintained a significant negative correlation with most cytokines, including *TNF*, *IL-6*, *CCL8*, and *IL-10*, but positive with *CSF1* and *TGFB2* (Fig. 4G). Unlike the early treatment-naive patients (Fig. 1E) and opposite to Axl behaviour, synovial MerTK showed a strong positive correlation with most cytokines/chemokines pathogenetically relevant in RA (Fig. 4G). In keeping with its regulatory role, lower levels of Axl synovial gene expression were significantly associated with higher CRP (Fig. 4H) but not with other clinical parameters, while no clinical correlation was observed for *MERTK*.

### Synovial Axl gene expression is modulated by the anti-IL-6 targeting treatment

To corroborate the known ability of TAM receptors to regulate inflammatory processes in RA, we next assessed the potential clinical implications of pre-treatment expression of Axl/MerTK in the synovium of patients in whom TNF inhibitors were ineffective about to receive either rituximab (RTX) or tocilizumab (TOC).

Notably, regression analysis revealed a significant differential *AXL/MERTK* correlation between responders and non-responders ($p_{interaction} = 0.018$) following IL-6 inhibition (TOC-treated) (Fig. 5A), whilst this difference was not observed in the RTX-treated arm

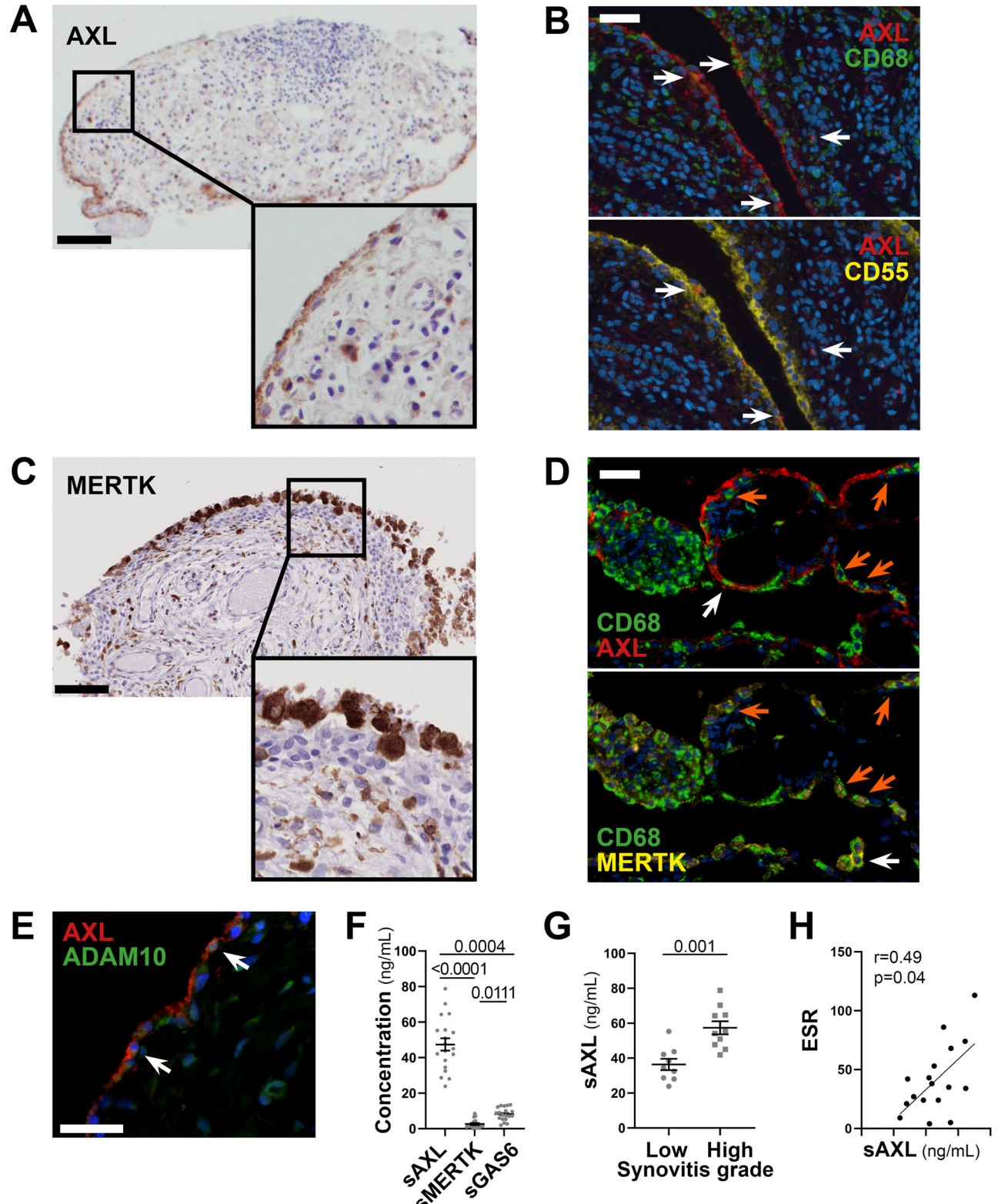

(p_interaction = 0.64), where *AXL* and *MERTK* were weakly correlated (Fig. 5B).

The close link between IL-6 and TAM receptors was also suggested by the significant upregulation of *AXL* synovial expression upon the blockade of IL-6 with TOC in both responders and non-responders (Fig. 5C). *AXL* significant enhancement was not observed upon B-cell depletion by the anti-CD20 RTX (Fig. 5C), while MERTK expression in this cohort was not significantly influenced by any of these treatments

(Fig. 5D). In line with these observations, the ratio *AXL/IL6* was significantly downregulated following treatment with TOC but not RTX (Fig. 5E).

We next assessed how treatment interventions regulated Axl and MerTK protein expression in CD68+ synovial macrophages in the TNFα inhibitors-ir cohort. A qualitative analysis of 24 pre-treatment (RTX n = 16, TOC n = 8) and 8 post-treatment (RTX n = 3, TOC n = 5) synovial tissues showed that following biologic therapies, all synovial

**Fig. 3 | Axl and MerTK have distinct and characteristic expression patterns in RA synovium, and Axl ectodomain can be cleaved and released in the synovial fluid. A, C** Representative images of Axl (**A**) and MerTK (**C**) immunohistochemistry (IHC) staining of synovial tissue sections. Scale bar = 100 µm. Representative images of *n* = 20 samples stained. **B** Double immunostaining of Axl (red) with CD68 (green, upper panel) and CD55 (yellow, lower panel) in the synovium of RA patients. Nuclei were counterstained with DAPI (blue). White arrows indicate double-positive cells. Scale bar = 50 µm. Representative images of *n* = 13 samples stained. **D** Triple immunostaining of Axl (red), MerTK (yellow), and CD68 (green) in the synovium of RA patients showing the presence of both Axl⁺ and MerTK⁺ double-positive CD68⁺ macrophages (orange arrow) and Axl⁺ or MerTK⁺ single positive CD68⁺ macrophages (white arrow). Nuclei were counterstained with DAPI (blue). Scale bar = 50 µm. Representative images of *n* = 18 samples stained. **E** Double

immunostaining of Axl (red) with ADAM10 (green) in the synovium of RA patients. Nuclei were counterstained with DAPI (blue). White arrows indicate double-positive cells. Scale bar = 50 µm. Representative images of *n* = 5 samples stained. **F** Levels of soluble Axl (sAxl), soluble MerTK (sMerTK) and soluble Gas6 (sGas6) in ng/mL assessed by ELISA in the synovial fluid of RA patients (*n* = 18). *p* values indicated were calculated using the Kruskall–Wallis test, with Dunn's post hoc test. **G** Levels of soluble Axl (sAxl) in ng/mL assessed by ELISA in the synovial fluid of RA patients (*n* = 18) divided according to synovitis score (low [0–4], high [5-9]). *p* values indicated were calculated using the two-tailed Mann–Whitney test. **F, G** Data are represented as mean ±SEM. **H** Correlation between sAxl synovial fluid levels and the erythrocyte sedimentation rate (ESR) of RA patients (*n* = 18). *p* value and *r* coefficient were calculated according to the two-tailed Spearman correlation test.

macrophages acquired Axl expression on their surface, which was expressed either alone or co-expressed with MerTK. Notably, about 75% of macrophages pre- and post-treatment were MerTK⁺ (Fig. 5F).

### Digital spatial profiling defines the positional identity of *AXL* and *MERTK* gene expression in rheumatoid synovium

Since positional identity has recently been reported to play an important role in the rheumatoid synovial tissue[3,17], we further investigated the molecular expression of *AXL*, *MERTK* and their gene partners in relation to their spatial distribution by applying digital spatial profiling (DSP) with the NanoString GeoMX DSP. ROIs selected for profiling consisted of lining, deep sublining and lymphocytic aggregates (Fig. 6A).

Both *AXL* and *MERTK* were upregulated in lining and sublining compared to lymphocytic aggregates (Supplementary Fig. 5), reflecting the lineage expression by macrophages and fibroblasts. In line with the protein data, and as shown in the polar plot (Fig. 6B), we demonstrated that *MERTK* gene expression was upregulated in the lining layer (Fig. 6C). On the other hand, *AXL* transcript was among the genes upregulated in both lining and sublining, suggesting the existence of post-translational mechanisms able to specifically downregulate its expression at the protein level in cell-subsets of the sublining (e.g., Thy-1+ FLS).

We next compared the expression of Axl, MerTK, and their gene partners in responders (to either RTX or TOC, *n* = 8) *versus* refractory patients (who failed both RTX *and* TOC, *n* = 4) in each synovial region. Among MerTK partners, *CD28* was upregulated in the sublining of responders, while *FCGR1A/CD64* and *c-MET* were significantly upregulated in refractory patients (sublining and lining, respectively). *ERRB2*, common to both Axl and MerTK modules, was also upregulated in sublining and aggregates of refractory patients (Fig. 6D). Even if the differential expression of *AXL* and *MERTK* did not reach statistical significance, taken together, these data further confirm that Axl and MerTK define distinct subsets of synovial cell populations and closely interact with biologic partners that are clinically relevant.

## Discussion

Our study provides insights into the biology and expression pattern of TAM receptors Axl and MerTK in RA in relation to disease activity and treatment exposure, adding data to the understanding of the role played by these tyrosine kinases in chronic inflammatory diseases.

Despite the management and prognosis of patients with RA have considerably improved over the last twenty years, ~40% of patients do not respond to individual medications and 10–20% are multi-drug resistant. In addition, the inability to predict therapeutic response and prevent radiologic progression in the erosive subgroup still leaves a huge unmet need and requires to better define pathogenetic mechanisms at individual patient level to find new therapeutic candidates. Axl and MerTK were proposed as chief regulators of inflammation soon after their discovery in animal models and, in recent years, specifically in synovial inflammation[4,10]. Here, we shed light on

their modulatory capacity in the context of RA clinical manifestations and response to therapy. We showed that higher levels of *AXL* synovial transcript are associated with lower immune-inflammatory infiltrate and less active disease, and negatively correlate with numerous pro-inflammatory cytokines in the synovium, independently of the disease stage or the exposure to treatment. This is in line with its role as a negative regulator of the inflammatory cascade, particularly the production of cytokines like TNFα and IL-6[18,19]. On the other hand, Axl displays a significant positive correlation with its known enhancer TGFβ[20], which has been shown to characterise low-inflamed rheumatoid synovitis[21]. Since both Axl and the negative regulator of inflammation Suppressor Of Signalling Cytokines (SOCS) 3 are induced by TGFβ[20], this axis may represent a key pathway exploitable to promote a low-inflammation status in the synovial tissue.

We found that in early RA, before treatment intervention, Axl protein is predominantly expressed by cells of the lining (either macrophages or fibroblasts), confirming in humans this preferred location previously suggested to be functional in animal models. Indeed, Culemann and colleagues[3] reported that Axl is expressed in the synovial lining and would functionally contribute to secluding the synovial space and protecting it from the influx of inflammatory arthritogenic molecules. Consistently, it has been proposed that the limited or absent Axl expression within the lining characterises joints that are more likely to be affected during inflammatory arthritis[2]. Since studies in healthy murine bone marrow-derived macrophages and DCs showed that Axl expression is significantly enhanced by pro-inflammatory stimuli like LPS and TNF[11], it was somehow surprising to note that, in early untreated active RA, most of the sublining cells, including both macrophages and CD90⁺ fibroblasts, do not express Axl on their surface; this was even further striking in the context of the DSP data showing that *AXL* transcript is among the genes upregulated in both lining and sublining. This is likely due to post-translational modifications linked to the RA tissue environment, including the cleavage of Axl ectodomain, as we demonstrated in this paper, and/or the epigenetic-driven down-regulation of Axl protein in cell-subsets of the sublining (e.g., Thy-1+ FLS or sublining macrophages), as previously demonstrated by Kurowska-Stolarska and colleagues in RA DCs[4].

Because of the ability of TAM receptors to be proteolytically cleaved, we sought to quantify sAxl ectodomain concentration in the SF of early untreated RA patients. In keeping with recent reports showing that Axl is one of the most abundant proteins in RA SF[6] and is elevated in RA *versus* OA[22], we demonstrated that sAxl is detectable in RA SF in excess to Gas6 and sMerTK, suggesting a role as a decoy for Gas6 due to Axl's high affinity for this ligand[23]. Also, in line with previous evidence indicating that Axl cleavage in vitro is induced by inflammatory stimuli[24], we confirmed a significantly increased shedding in the matched highly inflamed synovium. Therefore, in uncontrolled disease with high levels of inflammation, the continuous production of metalloproteinases would favour the perpetuation of inflammation by increasing the shedding of sAxl that, acting as a decoy

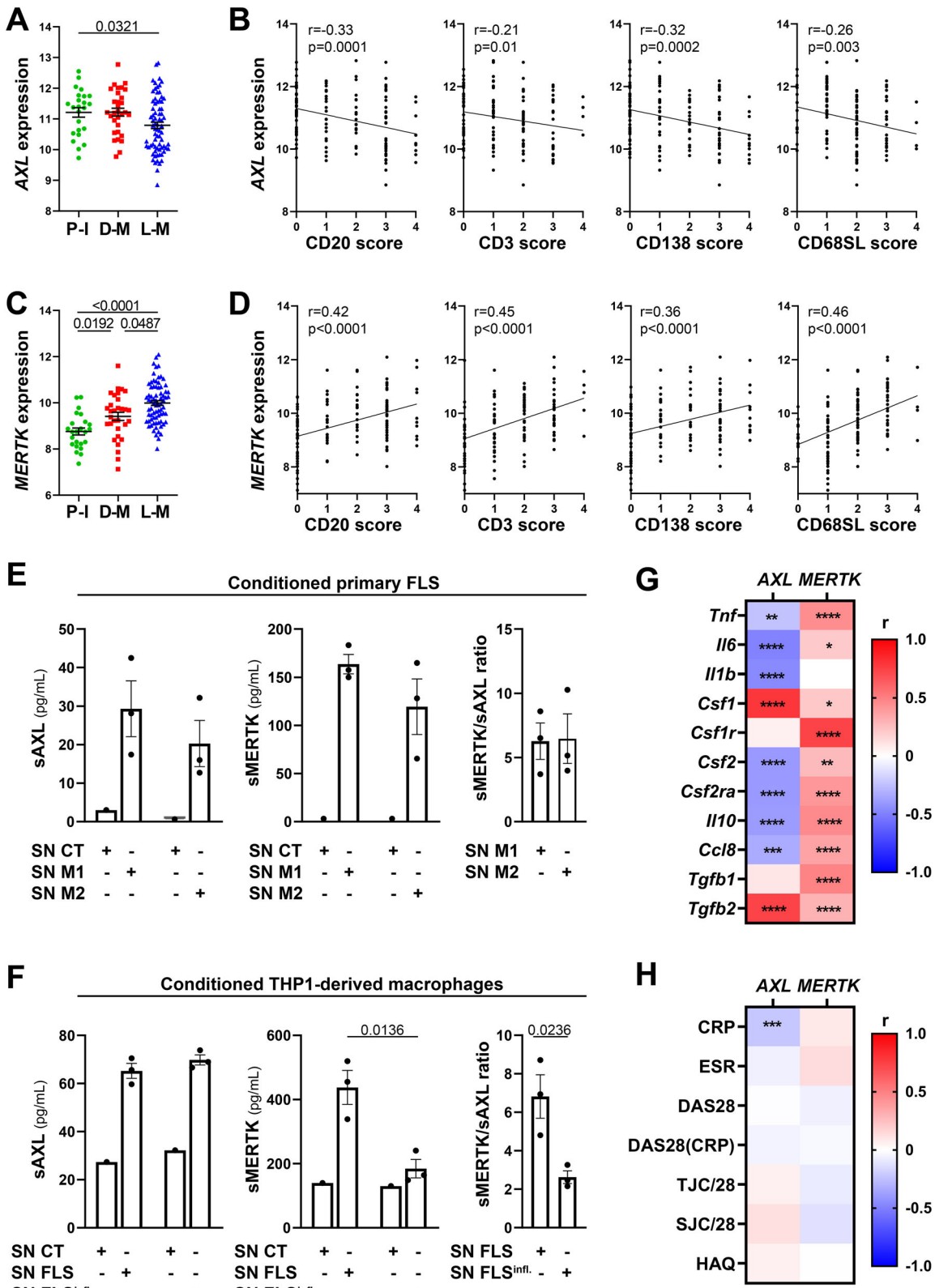

for Gas6, would prevent its binding with functional transmembrane Axl and MerTK, and preclude the anti-inflammatory activity of TAM receptors. Notably, an increased sAxl release has been recently proposed as a potential pathogenic mechanism in lupus[25]. Previous work on synovial micromasses containing FLS and macrophages demonstrated that TGF-β1 stimulation upregulates Axl[2]; here, we built in vitro systems to further characterise the interaction between stromal and myeloid cells and their mutual effects on driving TAM receptors shedding. We showed that both macrophage-like cells and primary RA-derived FLS release soluble Axl; in this model, Axl shedding is not significantly influenced by the macrophage polarisation status or additional TLR-driven activation of RA-derived FLS, overall suggesting a constitutive shedding of Axl ectodomain in the contest of RA. Taken together, our findings suggest that defective Axl synovial expression

**Fig. 4 | AXL and MERTK synovial expression and relationship with clinical features are influenced by the disease stage and treatment exposure. A, C** *AXL* (**A**) and *MERTK* (**C**) gene expression (vst normalised reads) in synovial tissue of anti-TNF inadequate responder RA patients (R4RA cohort, *n* = 133) according to the histological pathotype defined as Pauci-Immune (P-I, in green), Diffuse-Myeloid (D-M, in red), and Lympho-Myeloid (L-M, in blue). Data are represented as mean ±SEM. *p* values were calculated using the Kruskal–Wallis test with Dunn's post hoc test. **B, D** Correlation between synovial *AXL* (**B**) and *MERTK* (**D**) gene expression and semi-quantitative scores (0–4) of B cells (CD20), T cells (CD3), plasma cells (CD138), and sublining macrophages (CD68SL). *p* values and r-coefficients were calculated using the two-tailed Pearson correlation test. **E** Expression of sAxl, sMerTK (pg/mL) or the ratio between sMerTK and sAxl in the supernatant of primary fibroblasts-like synoviocytes (FLS) conditioned with supernatant from M1-polarised THP1 (SN M1) or M2-polarised THP1 (SN M2), or in the respective medium used to condition the cells (SN CT). **F** Expression of sAxl, sMerTK (pg/mL) or the ratio between sMerTK

and sAxl in the supernatant of THP1-derived macrophages conditioned with supernatant from unstimulated RA-FLS (SN FLS) or LPS stimulated FLS (SN FLS Infl.), or in the respective medium used to condition the cells (SN CT). **E, F** Data are represented as mean ±SEM. *p* values indicated were calculated using the unpaired two-tailed *t* test (left and middle panels) or the two-tailed Mann–Whitney test (right panel). Experiments were performed on *n* = 3 distinct patient-derived FLS. **G, H** Heatmaps showing the correlation between *AXL* and *MERTK* synovial transcript levels at baseline and cytokines and growth factors relevant to the inflammatory response (**G**) and clinical parameters (**H**). The red/blue scale represents the Spearman *r* coefficient, calculated using the two-tailed Spearman correlation test. *$p < 0.05$, **$p < 0.01$, ***$p < 0.001$, ****$p < 0.0001$. CRP C-reactive protein, ESR erythrocyte sedimentation rate, DAS28 disease activity score 28, TJC/28 tender joints count (0–28), SJC/28 swollen joints count (0-28), HAQ health assessment questionnaire.

and function in early active RA might contribute to the establishment and perpetuation of inflammation in the synovial tissue.

While in early treatment-naive patients, MerTK expression is not related to a specific pathotype, in established active RA, following treatment with TNF inhibitors and corticosteroids, MerTK gene is significantly upregulated in synovial tissues characterised by abundant immune cells (myeloid and lymphoid) infiltration, and it positively correlates with numerous pro-inflammatory cytokines. Interestingly, these findings differ from the AMP scRNA-seq[9] data showing that MerTK is more expressed in leucocyte-poor RA synovial tissue (Supplementary Fig. 6). This discrepancy might be explained by the different molecular analyses (scRNA-seq vs whole tissue RNA-seq) but also by distinctive features of our cohort of patients, all with active disease following treatment with TNF inhibitors, i.e., non-responder patients. Since the expansion of MerTK-positive synovial macrophages has been linked to the maintenance of clinical remission[10], the upregulation of MerTK in immune-rich and highly inflamed synovia could represent a counter-mechanism to repristinate tissue homoeostasis. We observed that conditioning FLS with macrophage-derived supernatants do not significantly influence the release of sMerTK, independently of their polarisation status. On the other hand, sMerTK shedding by macrophages is significantly reduced when cells are conditioned with TLR4-activated RA-derived fibroblasts. Previous collaborative work showed that macrophage membrane MerTK restrains pro-inflammatory effects of FLS, when co-cultured[10]. Here, we add the concept that TLR-activated FLS can indirectly affect the release of sMerTK by macrophages, favouring the hypothesis that MerTK (transmembrane)/ligand axis enhancement may counteract unrestrained inflammation.

Since the presence of a pauci-immune pathotype and a fibroblast-rich signature associate, respectively, with non-response to anti-TNF agents and refractory RA[16,26], MerTK upregulation might also be one of the factors contributing to the achievement of clinical response and, ultimately, remission, as supported by previous collaborative work[10]. The altered pattern of expression at this later stage of the disease compared to the early treatment-naive is also in line with data showing that both corticosteroids and TNF inhibitors enhance MerTK expression[8,27]. MerTK, differently from Axl, is mostly restricted to macrophages; interestingly, it is also expressed by tingible-body-macrophage-like cells within the lymphocytic aggregates, characterising lympho-myeloid patients express MerTK, suggesting an involvement in the clearance of apoptotic cells as per secondary lymphoid organs[11].

The diverse expression of Axl and MerTK by distinct macrophage subsets has been previously elegantly demonstrated in ex-vivo experiments using murine myeloid cells. For instance, while alveolar macrophages are exclusively Axl-positive, tingible-body macrophages in secondary lymphoid organs are solely MerTK-positive, and hepatic Kupffer cells display dual Axl/MerTK expression[11,28]. Here, we show

that, in rheumatoid synovial tissue, distinct macrophage subsets, either single Axl+, MerTK+ or double Axl+MerTK+, are present and dynamically regulated by the positional identity of macrophages and specific therapeutic interventions. Indeed, upon blocking the IL-6 pathway by administering the IL-6 receptor inhibitor tocilizumab, *AXL* synovial transcript levels become significantly upregulated, the ratio between Axl and IL-6 post-treatment is significantly reduced, and most synovial macrophages acquire Axl on their surface, suggesting that IL-6 pathway modulation can affect Axl expression. In line with these findings, in both early arthritis and established RA cohorts, Axl is consistently inversely correlated with IL-6 and CRP; moreover, we showed that the pathway analysis of the Axl module, including Axl gene-partners, is specifically enriched with IL-6-mediated signalling and cellular response to IL-6 pathways. Although the upregulation of Axl following IL-6 blockade was observed in both responders and non-responders at 16-week, we show that the correlation between Axl and MerTK is significantly different between responders and non-responders to tocilizumab, with patients achieving a good response to IL-6 inhibition (but not to B-cell depletion with rituximab) being characterised by a positive correlation between the two TAM receptors. Altogether, our findings suggest that both synovial Axl and MerTK are influenced by exposure to treatment and linked to synovial pathology; we also propose a regulatory mechanism for Axl synovial expression in RA driven by IL-6 receptor inhibition therapy in vivo.

To gain further knowledge of the upstream and downstream pathways linked to Axl and MerTK, we built an Axl and a MerTK module composed of predicted gene partners and assessed their expression in the early arthritis RNA-seq dataset.

We show that, among Axl partners, genes such as *IGF1-R* and epidermal-growth factors receptors (*ERBB2, EGFR*), previously linked to proliferation and survival of synoviocytes in RA[29], are more expressed in pauci-immune pathotypes, rich in stromal cells; moreover, *ERBB2* is significantly upregulated in the synovial sublining of patients who are refractory to multiple therapeutic agents. Axl itself is mitogenic and anti-apoptotic in fibroblasts[23], but it exerts its anti-inflammatory activity in myeloid antigen-presenting cells via type I interferon signalling inhibition. Accordingly, Axl-partner *IFNAR1*, encoding for Interferon α- and-β-Receptor-Subunit1, appears to be upregulated in the lympho-myeloid pathotype. Though additional functional studies are required, based on our findings, we suggest that, within the rheumatoid synovium, Axl may play alternative roles in distinct cell lineages/pathotypes, thus influencing the modulation of inflammation, the disease progression, and the response to treatments.

Axl module also includes several genes of the PI3K family, which, similarly to Axl[30], are positive regulators of vascular endothelial growth factor (VEGF) and effective mediators of the neovascularisation of the synovium in animal models of RA[31]. Axl role in angiogenesis is also further supported by pathway analysis showing enrichment of VEGF signalling and the positive regulation of angiogenesis.

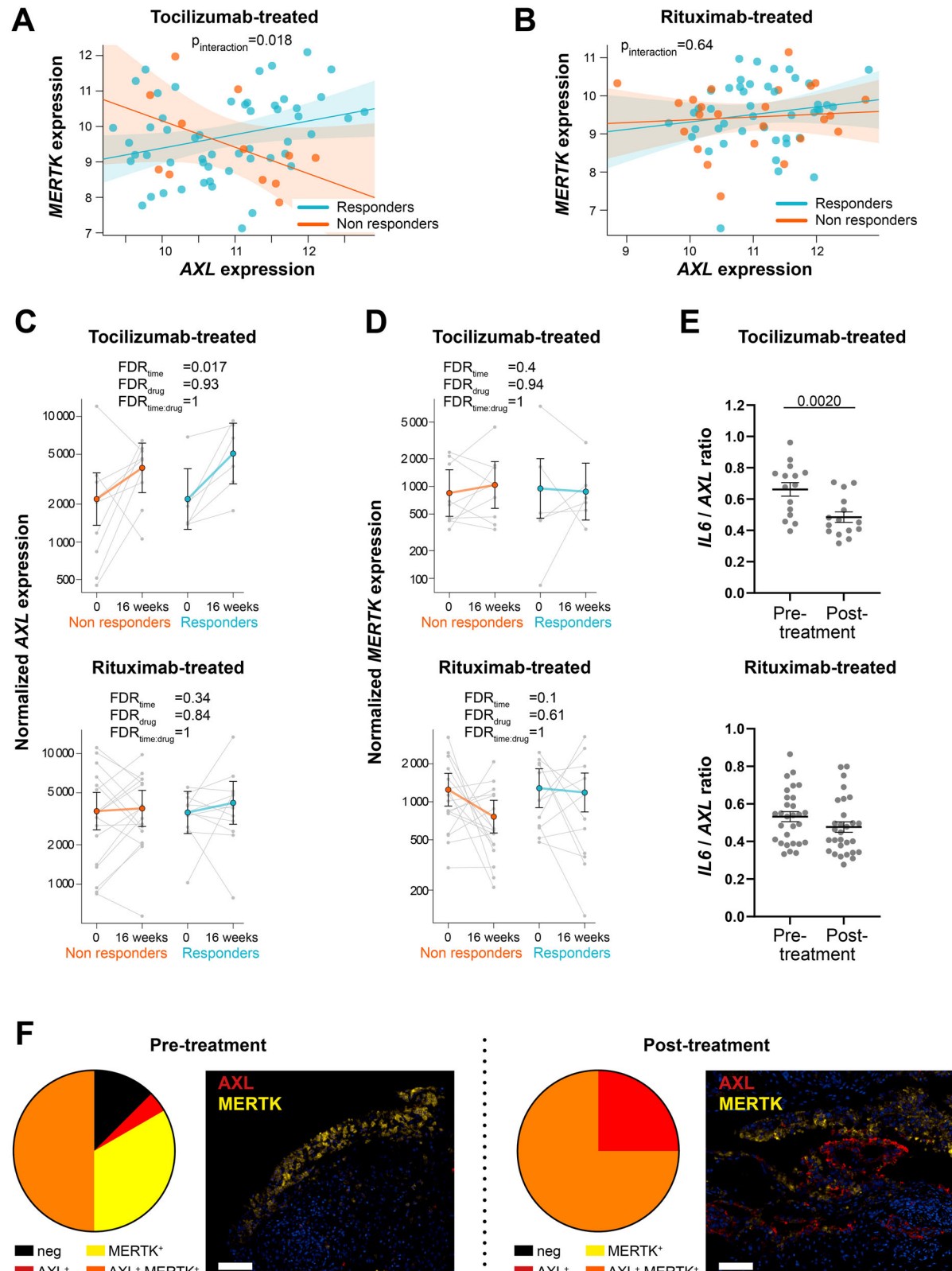

MerTK module was instead specifically characterised by several gene partners encoding proteins that play essential roles in macrophages. *PTPN1* (or *PTP1B*), for example, known to negatively regulate macrophage development and activation through CSF1 signalling[32], is upregulated in lympho-myeloid RA patients; similarly, *CD28*, encoding for the CD80/CD86 receptor and fundamental for T cells/antigen-presenting-cell interaction, is upregulated in lympho-myeloid and the sublining of patients responding to biologic treatments. Also, MerTK partners *CD64/FCGR1A*, encoding for Fc-gamma receptor 1A, and *c-MET* emerged to be clinically relevant. The former, upregulated in lympho-myeloid, is significantly higher in the sublining of patients refractory to treatments[33], in line with previous works showing that a reduction of CD64[33] accompanies favourable responses to anti-rheumatic drugs; the latter, more expressed in pauci-immune and

**Fig. 5 | *AXL* is modulated by IL-6 pathway inhibition. A**, **B** Linear regression model analysis with interaction term to estimate the correlation of *AXL* with *MERTK* expression in relation to clinical response to tocilizumab (**A**) and rituximab (**B**) in anti-TNF inadequate responder RA patients (R4RA cohort, *n* = 133, including 65 tocilizumab-treated and 68 rituximab-treated patients). The clinical response was assessed by EULAR criteria with DAS28(CRP) after 16 weeks of treatment (good responders in light blue; moderate and non-responders in orange). p-interaction is significant (0.018) in the tocilizumab-treated patient group and not significant in the rituximab-treated patient group. The scatter plots show the regression line of the fitted negative binomial generalised mixed effects model with the error bars showing 95% confidence interval (fixed effects). **C**, **D** *AXL* (**C**) and *MERTK* (**D**) normalised gene expression levels assessed at baseline and 16 weeks following tocilizumab (left panels, *n* = 15 matched samples) or rituximab (right panels, *n* = 29

matched samples) treatment. CDAI 50% improvement was used to assess the clinical response (responders in light blue, non-responders in orange). Statistical analysis was performed by negative binomial generalised mixed effects model. FDR: false discovery rate. Data are shown as mean ±95% confidence interval. **E** *IL-6/AXL* expression ratio in the synovial tissue of tocilizumab- (*n* = 15) or rituximab- (*n* = 29) treated patients. Data are represented as mean ±SEM. *p* values indicated were calculated using the two-tailed Wilcoxon test for paired data. **F** Pie chart showing the percentage of Axl or MerTK single positive, MerTK and Axl double-positive and negative synovial tissue of anti-TNF inadequate responder RA patients (R4RA cohort) at baseline pre- (*n* = 24) and post-treatment with either tocilizumab or rituximab (RTX-treated *n* = 3, TOC-treated *n* = 5) (left panels) and representative images of double immunofluorescence staining for Axl (red) and MerTK (yellow) (right panels). Scale bar = 50 μm.

diffuse-myeloid patients, is likewise upregulated in the lining of refractory patients. Interestingly, NK4, a c-MET antagonist, has been shown to ameliorate arthritis[34]. Overall, these data indicate that Axl, MerTK and their gene partners are heterogeneously expressed in the tissue and linked to diverse clinical responses, thus representing possible candidates for therapeutic targeting in non-responder patients.

The findings of our study must be considered in the context of some limitations. First, most of the molecular data about Axl and MerTK derive from bulk-RNA-seq. While this is truly representative of the overall tissue signals, it is not possible to appreciate the distinct contribution of single-cell types (e.g., macrophages, DCs, fibroblasts) to MerTK and, especially, Axl expression. Second, the longitudinal variation of Axl and MerTK across disease stages and treatment exposure have been assessed at cohort level and not at individual patient's level; nonetheless, our investigation in unique large cohorts of patients synchronised for disease stage and treatment exposure ensures high-quality data and robust findings. Third, although by the same nature of being an ex-vivo study the evidence of an IL-6-driven Axl modulation is indirect, the modulation post-tocilizumab but not post-rituximab therapy is strong evidence in favour of this mechanism.

In conclusion, we provided evidence in the clinical context that MerTK and Axl are pleiotropic receptors capable of mediating multiple biological functions with either overlapping or distinct expression patterns and mechanistic profiles. While they are both involved in downregulating the inflammatory response, Axl is also pro-angiogenic and anti-apoptotic, whereas MerTK is critically involved in the clearance of apoptotic cells. In our in vitro model, we demonstrated that activated RA-FLS can modulate MerTK release by macrophage-like cells, proving that different cell types and activation statuses influence the overall expression and function of the TAM axis. Our ex-vivo patient data further corroborate that Axl and MerTK constitute a dynamic axis in RA, influenced by the synovial tissue inflammatory features, the disease stage, the exposure and the response to targeted treatments and the blockade of critical inflammatory pathways over time. Since the overall effect of TAM receptors activation in animal models is the amelioration of arthritis[35], a better understanding of how individual features of these tyrosine kinases are modulated in the inflammatory environment of the RA joint, as well as their interaction, would help the definition of potential treatment approaches and mechanisms/targets for patients not responding to current medications.

## Methods

### Patient cohorts

**Pathobiology of Early Arthritis Cohort (PEAC).** The PEAC (http://www.peac-mrc.mds.qmul.ac.uk) is an observational study, which started in 2008 and is still open to recruitment. This cohort is well-established and described in several publications[12,36]. Following written informed consent, patients undergo ultrasound (US)-guided synovial biopsy of their most inflamed joint[37]. When available, SF is also collected prior to the procedure. An optional post-treatment biopsy of

the same joint is repeated 6-months after starting the treatment. In this manuscript, we analysed data from the first sequential RA patients, who had been RNA-sequenced. All patients included (*n* = 87, of whom 73.6% female, for RNA-seq analysis; *n* = 18, of whom 72.2 % female, for SF analysis) were treatment-naïve for both Disease-Modifying-Anti-Rheumatic-Drugs (DMARDs) and steroids, had a disease duration <12 months and fulfilled the ACR/EULAR 2010 classification criteria for RA. Following the baseline biopsy, patients are commenced on conventional synthetic (cs) DMARDs and/or low-dose corticosteroids as per standard of care according to the UK National Institute for Clinical Excellence prescribing algorithm. Patients' sex was collected from medical records. The study was approved by the National Research Ethics Service Committee London Dulwich (REC 05/Q0703/198).

**Anti-TNF inadequate responder patients cohort (R4RA cohort).** The Rituximab versus tocilizumab in anti-TNF inadequate responder (ir) patients with rheumatoid arthritis (R4RA) was the first multi-centre synovial biopsy-driven randomised clinical trial recruiting difficult-to-treat RA patients between 2013 and 2019, as described in Humby et al.[15] and Rivellese F et al.[16]. Upon providing written informed consent, a total of 161 patients (80% female) fulfilling the 2010 ACR/EULAR classification criteria for RA who failed the first-line biologic treatment with Tumour Necrosis Factor (TNF) inhibitors underwent a baseline US-guided or arthroscopic synovial biopsy before being randomised to receive either rituximab (RTX) or intravenous tocilizumab (TOC). An optional repeated synovial biopsy of the same joint sampled at baseline was performed at 16 weeks in a subset of patients, when the clinical response was also recorded, and patients switched accordingly. The study protocol, including baseline and optional repeated synovial biopsy, was approved by the institutional review board of each study centre or relevant independent ethics committees (UK Medical Research and Ethics Committee (MREC) reference12:/WA/0307). Patients' sex was collected from medical records. Axl and MerTK molecular expression was assessed at baseline in *n* = 133 samples, including *n* = 68 subsequently treated with RTX, and *n* = 65 subsequently treated with TOC. Paired analysis (Fig. 5C, D) was performed on *n* = 29 patients treated with RTX and *n* = 15 treated with TOC. Protein expression analysis was performed in *n* = 24 (baseline samples) and *n* = 8 (16-week samples).

### Immunostaining and histological analysis

Formalin-fixed paraffin-embedded (FFPE) 3 μm-thick synovial tissue sections were stained by haematoxylin and eosin (H&E) to determine the synovitis score, which was defined using the previously published Krenn's score (0-9)[38]. After confirming the presence of the synovial lining, immunohistochemistry (IHC) with antibodies specific for CD68 (Dako, M0814, KP1, Mouse IgG, 1:50, Cat. N. GA609), CD3 (Dako, M7254, F7.2.38, Mouse IgG1 kappa, 1:50, Cat. N. M7254), CD20 (Dako, M0755, L26, Mouse IgG2a kappa, 1:50, Cat. N. M0755) and CD138 (Dako, M7228, MI15, Mouse IgG1 kappa, 1:50, Cat. M7228) (all from Agilent Technologies, USA) was performed to determine the synovial

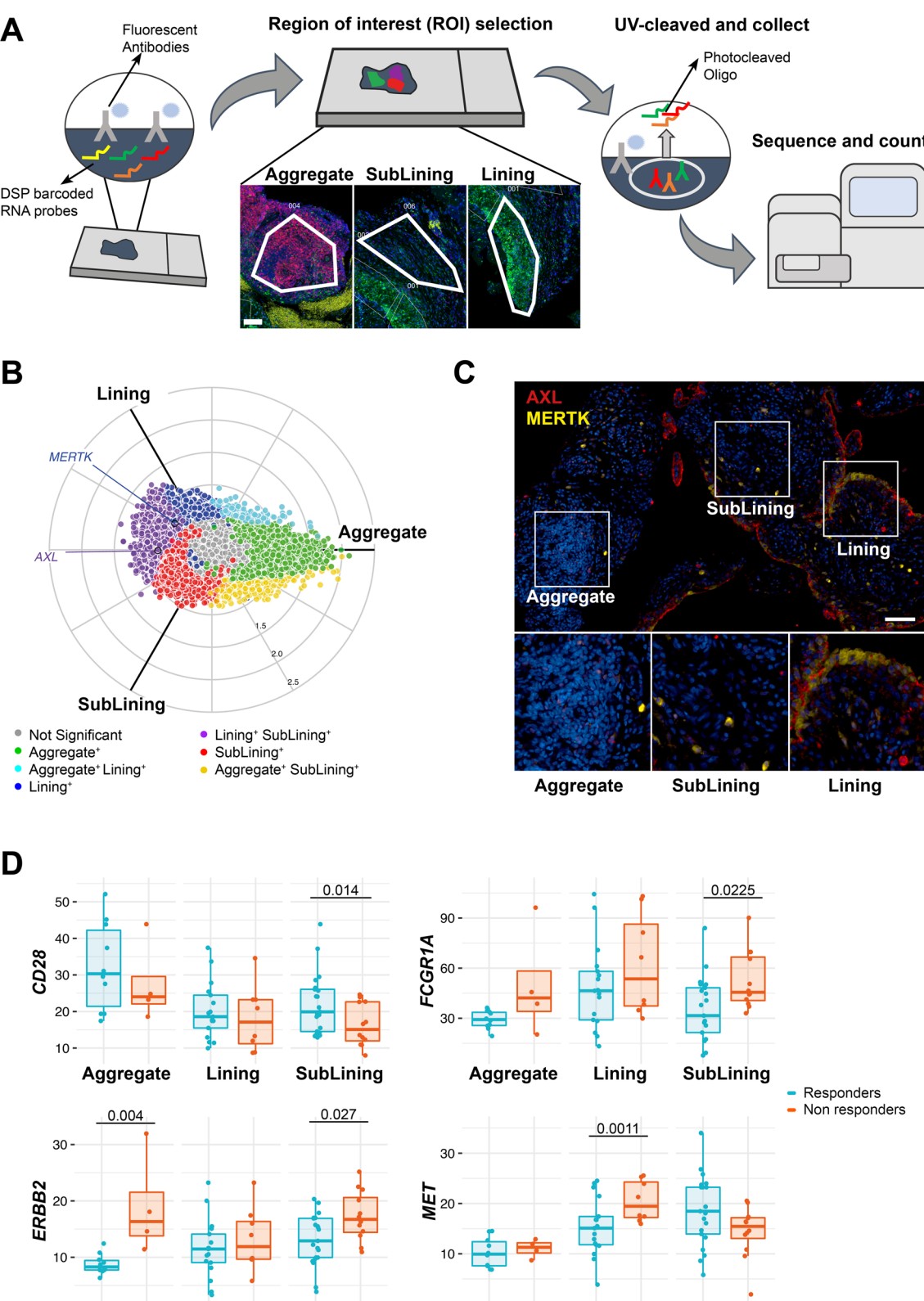

pathotype (pauci-immune/fibroid, diffuse-myeloid or lympho-myeloid) following semi-quantitative scoring as described previously[36]. EnVision+ System-HRP labelled anti-mouse (Dako, K4001, Polyclonal, Goat anti-mouse) was used as the secondary antibody. PEAC semi-quantitative immuno-scores and pathotype classification were performed by two researchers expert in synovial pathology at Queen Mary University of London (QMUL). R4RA patients were initially scored in the pathology laboratory of Barts Health NHS Trust (London, UK) by a consultant pathologist; histological scores were replicated at QMUL by an independent expert. In both cases, discrepancies in classification were resolved through mutual agreement. Slides were also stained for TAM receptors tyrosine kinase Axl (R&D Systems, USA, AF154, Polyclonal, Goat IgG, 1:200) and MerTK (Abcam, USA, ab52968, Y323, Rabbit IgG, 1:500); the following secondary antibodies were used,

**Fig. 6 | Digital spatial profiling (DSP) confirms the positional identity of *AXL* and *MERTK*. A** Schematic representation of the Digital Spatial Profiling (DSP) approach, including the selection of three regions of interest (ROI): aggregate (characterised by the presence of CD3+ and CD20+ cells), deep sublining (characterised by the absence of CD3+ and CD20+ cells) and lining with superficial sublining (characterised by the presence of CD68+ cells). Scale bar = 100 μm. DSP was performed on 14 aggregates, 25 lining, and 33 sublining regions. **B** Three-way radial plot showing differential and overlapping genes across aggregate (green), lining (blue) and sublining (red) regions. *AXL* and *MERTK* genes have been labelled showing a significantly higher presence of *AXL* in both lining and sublining regions and a significant presence of *MERTK* in the lining region. Significance was internally estimated by the volcano3D package combining significance ($q < 0.05$) from both

one-way ANOVA and pairwise *T* test. **C** Double immunostaining of Axl (red) and MerTK (yellow) in the aggregate, sublining and lining synovial areas of RA patients. Nuclei were counterstained with DAPI (blue). Scale bar = 50 μm. Representative images of $n = 32$ samples stained. **D** Expression of selected individual genes included in the *AXL* and/or *MERTK* networks in the aggregate ($n = 14$, including 4 non-responder and 10 responder patients), sublining ($n = 33$, including 12 non-responder and 21 responder patients) and lining ($n = 25$, including 8 non-responder and 17 responder patients) synovial areas of responders (light blue) and non-responders (orange) RA patients to either tocilizumab or rituximab. Boxplots represent the median and first and third quartiles, and whiskers span to the minimum and maximum. A paired Wilcoxon test was undertaken to compare responders and non-responders.

respectively: VisUCyteTM anti-Goat (R&D Systems, VC004-025, Polyclonal, Donkey anti-Goat IgG, ready-to-use) and Envision+ System-HRP labelled anti-rabbit (Dako, K4003, Polyclonal, Goat anti-rabbit, ready-to-use). Slides were counterstained with haematoxylin and mounted with Distyrene Plasticizer Xylene (DPX) mounting medium (Cat. N. 06522, Sigma-Aldrich, USA). All sections were digitally scanned using Nanozoomer S210 (Hamamatsu Photonics, Japan).

Double and triple fluorescent labelling was performed on synovial sections by multiplex immunofluorescence staining using a tyramide signal amplification (TSA) protocol (Invitrogen, Thermo Fisher Scientific, USA). The following primary antibodies were used in various combinations to assess co-localisation: CD68 (Dako, M0814, KP1, Mouse IgG, 1:50, Cat. N. GA609), Axl (R&D Systems, USA, AF154, Polyclonal, Goat IgG, 1:200), ADAM10 (Abcam, ab124695, EPR5622, Rabbit IgG, 1:100), MerTK (Abcam, USA, ab52968, Y323, Rabbit IgG, 1:500), CD55 (Abcam, ab133684, EPR6689, Rabbit IgG, 1:100) and CD90 (Abcam, ab133350, EPR3133, Rabbit IgG, 1:240). The following secondary antibodies were used at 1:200 dilution: AlexaFluor488 (B40953), AlexaFluor555 (B40955), and AlexaFluor647 (B40958), all from Invitrogen, ThermoFisher Scientific. Slides were counterstained with 40,6-diamidino-2-phenylindole (DAPI, Cat. N. D1306) and mounted with ProLong Antifade Mountant (Cat. N. P36970), both from Invitrogen, Thermo Fisher Scientific, USA. Images were captured using a Nanozoomer S60 Digital slide scanner (Hamamatsu Photonics, Japan) and visualised with NDP.view 2 Software (Hamamatsu Photonics, Japan). Quantitative digital image analyses were performed to determine the percentage of Axl, MerTK or CD68 positive cells within the synovial tissue stained by IHC or immunofluorescence using QuPath software (version 0.3.0)[39].

### THP1-derived-macrophages conditioned with RA-derived FLS media

To prepare the conditioning media, $n = 3$ RA-derived primary FLS (isolated from either synovial tissue or synovial fluid of an actively inflamed joint, $n = 2$ male and $n = 1$ female) were plated at confluency >90% in 48-well plates in Dulbecco's Modified Eagle Medium/Nutrient Mixture F-12 (Cat. N. 11320033, DMEMF12, ThermoFisher Scientific) and left resting overnight. Only cells between P3 and P4 were used for experiments. Cells were then stimulated with 100 ng/mL of lipopolysaccharide (LPS, Ultrapure from *E. coli* 0111:B4, tlrl-3pelps, InvivoGen) or vehicle (Supplementary Fig. 4A). Media from both conditions were collected after 48 hours and stored until used to stimulate THP1 cells. THP1 cells were plated at $0.5 \times 10^6$/mL in DMEMF12 and cultured with 10 ng/mL of 12-OTetradecanoylphorbol-13-acetate (PMA, Cat. N. 16561-29-8, Sigma) for 24h. Cells were then washed with phosphate-buffer-saline (PBS, Cat. N. 10010023, Gibco) and cultured with FLS-derived conditioning media (1:1) for further 24h when media were collected for analysis.

### Primary RA-derived FLS conditioned with THP1-derived macrophage media

To prepare the conditioning media, THP1 monocyte-like cells were plated at $0.5 \times 10^6$/mL in DMEMF12 and cultured with 10 ng/mL of PMA

for 24 h. After changing the medium (PMA removed), cells were left to rest for 72h. Polarisation stimuli were then added as follows: 100 ng/mL of lipopolysaccharide (LPS, Invivogen) and 20 ng/mL of interferon-gamma (IFNg, Cat. N. 300-02 Peprotech) for M1-polarisation; 100 ng/mL Interleukin-4 (IL-4, Cat. N. 200-04, Peprotech) for M2-polarisation[40] (Supplementary Fig. 4A). After 24h with polarising stimuli, cells were washed, and fresh medium was added. Media were then collected after 48 hours and stored at −80 °C until used to condition FLS. FLS were plated at confluency >90% in 48-well plates in DMEMF12 and left resting overnight. Conditioning media from polarised THP1-derived macrophages were added to FLS (1:2 in fresh DMEMF12). Media from conditioned FLS were collected after 48 hours for analysis.

### Soluble Axl, soluble MerTK, and Gas6 quantification in synovial fluid or cell supernatants

Soluble (s)Axl, sMerTK, and Growth-Arrest-Specific (Gas) 6 were quantified in synovial fluid samples collected at the time of the baseline synovial biopsy from early treatment-naive patients enrolled in the PEAC study using commercial ELISA Kits (respectively, R&D Systems DY154, BMS2285, Invitrogen, and R&D Systems DY885) following the manufacturer's instructions. sAxl, sMerTK, and Gas6 were quantified in the supernatants of THP1-derived macrophages and FLS using the following commercial kits: sAxl (EHAXL, Invitrogen), sMerTK (Cat. N. DY6488), and Gas6 (Cat. N. DY885B), both from R&D Systems, following the manufacturer's instructions.

### RNA extraction and sequencing

Total RNA was extracted from RA synovial tissues using either a Phenol/Chloroform method or Zymo Direct-zol RNA MicroPrep–Total RNA/miRNA Extraction kit (Cat. N. R2052, Zymo Research). Quality control was performed via spectrophotometer quantification on a Nanodrop ND2000C, and RNA integrity number (RIN) was determined using Pico-chip on an Agilent 2100 Bioanalyzer.

When available, 1 μg total RNA was used for library preparation using either TruSeq RNA Sample Preparation Kit v2 for Illumina (PEAC) or NEBNext Ultra RNA Library Prep kit for Illumina (R4RA cohort). The libraries were multiplexed and then sequenced on an Illumina HiSeq instrument as per manufacturer's instructions. Raw data quality control, normalisation, alignment, and analysis of normalised read counts were performed as previously described in refs. 12,16. PEAC RNA-seq data were uploaded to ArrayExpress and are accessible via accession E-MTAB-6141[12]; R4RA RNA-seq data can be downloaded from https://www.ebi.ac.uk/arrayexpress/experiments/E-MTAB-11611[16]. Statistical analysis of RNA-seq count data was performed using Bioconductor package DESeq2. Longitudinal analysis of RNA-Seq count data was performed by negative binomial generalised mixed effects model using R package glmmSeq, as described in depth in ref. 12.

### Axl and MerTK module generation and analysis

Module genes were selected by creating STRING networks[41] centred on the target gene and extending until >31 genes were included in

the network. Module scores for RNA-seq data were derived by singular value decomposition (SVD) for each gene module matrix using a methodology described in detail by other studies[42]. STRING homo sapiens database was accessed via https://string-db.org/ on 16th August 2021.

## Gene set enrichment analysis

GSEA was performed using the R interface to the EnrichR database (v3.0.0) (https://CRAN.R-project.org/package=enrichR).

## Axl/MerTK interaction analysis

To estimate the correlation of Axl with MerTK in relation to response, a robust linear regression model with interaction term was fitted using the rlm function from the MASS (v.7.3) R package with the form: $GeneA_i = \beta_0 + \beta_1\ GeneB_i + \beta_2\ Response_i + \beta_3\ GeneB_i\ *\ Response_i + \varepsilon_i$ where $i = 1, \ldots n$, is the number of samples and $\varepsilon_i$ are random variables. The p-value associated with the $GeneB_i * Response_i$ interaction term was observed to assess a statistically significant difference in correlation.

## NanoString GeoMx DSP

FFPE synovial tissue sections from 12 RA patients recruited into the R4RA clinical trial were profiled using the NanoString GeoMx DSP Whole Transcriptome Atlas (WTA) assay as previously described[16,43]. The following fluorescent markers were used to determine the morphology of the tissue: CD68-AF532 (clone KP1, Cat. N. NB100-683 Novus) for macrophages, CD20-DL594 (clone IGEL/773, Cat. N. NBP2-44745, Novus) for B cells, and CD3-AF647 (clone UMAB54, Cat. N. TA807198, Origene) for T cells; nuclei were counterstained with Syto13 (Cat. N. S7575, ThermoFisher). Slides were prepared as per the automated Leica Bond RNA Slide Preparation Protocol (NanoString, MAN-10131-03). In situ hybridisations with the NanoString GeoMx WTA panel, including 18,677 genes, were done in Buffer R (NanoString) at a 4 nM final concentration. Slides were then incubated with 125 µl of morphology marker solution at RT for 1 h, washed, loaded onto the DSP instrument, and scanned with a ×20 objective (FITC/525 nm, 60ms; Cy3/568 nm, 200 ms; Texas Red/615 nm, 250 ms; Cy5/666 nm, 300 ms).

Freeform polygon-shaped regions of interest (ROI) containing approximately 200 nuclei were selected remotely on immunofluorescent-stained images to include i. synovial tissue lining and superficial sublining (CD68+), ii. lymphocyte aggregates (CD20+CD3+), and iii. deep sublining (CD20−CD3−). Subsequently, individual segmented areas within ROIs were photocleaved by the NanoString GeoMx DSP and collected into separate wells of a 96-well collection plate. The final dataset generated included 72 ROIs from 4 refractory and 8 responder patients. An NTC water well was used for quality control checks.

NanoString GeoMx WTA sequencing reads were compiled into FASTQ files corresponding to each ROI. FASTQ files were converted to digital count conversion files using the NanoString NanoString GeoMx NGS DnD Pipeline. Q3 normalisation was performed as described in Rivellese et al.[16]. The PCA of the Q3 normalised data is shown in Supplementary Fig. 7. A mixed effects model was fitted via lme4 (v1.1-27.1) using the ROI location as fixed effects and patient IDs as random effects. One-way ANOVA and pairwise tests between ROIs were derived via the car package (v3.0-12) and used to show differential and overlapping genes across aggregate, lining and sublining regions in a three-way radial plot made via volcano3D (v1.3.1). Normalised expression of selected individual genes was represented in boxplots using ggplot2 (v3.3.6). Significance values from paired Wilcoxon tests were reported. Principal component analysis (PCA) on the normalised data was undertaken using prcomp from the stats package (v4.2.0) and plotted using ggplot2 (v3.3.6).

## Statistical analysis

Comparisons of continuous variables were analysed by Mann–Whitney $U$ test (unpaired samples, between two groups), Wilcoxon signed-rank test (longitudinal paired data) or Kruskal-Wallis with Dunn's post hoc test (multiple groups). Associations of categorical variables were assessed by Chi-squared or Fisher's exact tests, as required. Correlations were evaluated by Spearman's bivariate analysis or two-tailed Pearson correlation test, as required. Patients' related analyses were generated using individual values (one value for each patient); in vitro experiments were performed on $n = 3$ distinct patient-derived FLS. Data are shown as the mean of three technical replicates for each individual patient unless otherwise specified. Statistical tests applied and the exact $p$ values are detailed in the Figures and captions. Specific RNA-seq and NanoString GeoMx data analyses are detailed above. Statistical analyses were performed using GraphPad Prism-v9 software (GraphPad, San Diego, CA, USA) or R v.4.0.0. $p$ values < 0.05 were considered significant.

## Reporting summary

Further information on research design is available in the Nature Portfolio Reporting Summary linked to this article.

## Data availability

The PEAC data used in this study have been deposited in the ArrayExpress database under accession code E-MTAB-6141. The R4RA data used in this study have been deposited in the ArrayExpress database under accession code MTAB-11611. The processed PEAC data are available on https://peac.hpc.qmul.ac.uk/ as an interactive web interface that allows direct data exploration as published by Lewis et al.[12]. The processed R4RA data are available on https://r4ra.hpc.qmul.ac.uk/ as interactive web interface that allow direct data exploration as published by Rivellese F at al.[16]. The STRING homo sapiens database was accessed via https://string-db.org/ on 16th August 2021. Source data are provided with this paper.

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

## Acknowledgements

We thank all patients who were enroled in PEAC and R4RA, the Patient Advisory Group, the clinical staff who helped with recruitment and data collection, and the laboratory staff who helped with processing the synovial samples. We would like to thank all R4RA investigators and recruitment centres (http://www.r4ra-nihr.whri.qmul.ac.uk/recruiting_centres.php) and R4RA team at Queen Mary (http://www.r4ra-nihr.whri.qmul.ac.uk/docs/contributors_r4ra-for_website.pdf). The PEAC study was supported by funding from the UK Medical Research Council (MRC) [grant number G0800648]. The R4RA trial was funded by the Efficacy and Mechanism Evaluation (EME) Programme [grant number 11/100/76] and further supported by NIHR [grant number 131575] and MRC [grant number MR/V012509/1]. AN's clinical lectureship was supported by Versus Arthritis [grant number 21890]. MAB was funded by the "Fondation pour la recherche médicale" [grant number ARF202004011786] and by the Inserm "ATIP-Avenir" programme. FR was supported by an NIHR fellowship [grant number TRF-2018-11-ST2-002]. Core work associated with this project was supported by grants from Versus Arthritis [Experimental Arthritis Treatment Centre, grant number 20022]. This work acknowledges the support of the National Institute for Health and Care Research Barts Biomedical Research Centre (NIHR203330), a

delivery partnership of Barts Health NHS Trust, Queen Mary University of London, St George's University Hospitals NHS Foundation Trust and St George's University of London. Some illustrations were created using Servier medical art [smart.servier.com] under Creative Commons Attribution 3.0 Unported License.

## Author contributions

C.P. conceived and sought funding for PEAC and R4RA. A.N. is the first author in the list of shared first authors as she conceived this study, provided her expertise on TAM receptors, and initiated and supervised the overall project. M.A.B. is the second on the list, as she supervised the in vitro experiments and figure preparation and wrote the first draft of the manuscript with A.N. G.M.G. is the third co-first author, as she performed and guided most of the experimental work, conceived the in vitro models and performed all related experiments. C.P., M.J.L., M.Bo., F.R., E.P., E.Pr., K.G., and E.S. revised the manuscript. F.R., F.H., and M.Bo. were responsible for patients' recruitment and collection of samples. K.G., E.S., C.C., and R.L. performed the bioinformatic analysis; M.J.L. supervised the bioinformatic analysis. G.M.G., M.C., and M.B. performed the histology and immunofluorescence experiments. G.M.G. analysed the histology data. E.P. and M.A.B. guided the analysis of the histological data. S.P., D.M., F.A., and G.M.G. performed and analysed the synovial fluid experiments. F.A. and G.M.G. ran all the ELISAs. S.E.C. and B.M.H. ran the NanoString GeoMx spatial profiling analysis and advised on data analysis. All authors contributed to the discussion and interpretation of the results, critically reviewed the manuscript, and approved the final version for submission.
