## [Peer Review File · Nature Communications]

Axl and MerTK regulate synovial inflammation and are modulated by IL-6 inhibition in rheumatoid arthritisREVIEWER COMMENTS

Reviewer #1 (expert in interleukin-6, immunoregulation and autoimmunity):

The authors using R4RA biopsies shows a potential mechanism of how TAM receptors- AXL and MERTK- can modulate the joint inflammation and response to IL-6 blockade. However, the concept of the synovial AXL expression in modulating the inflammation in mouse and human joints has been addressed including the role of synovial fibroblasts and macrophages (Claire E. et al, Rheumatology 2019;58:536-546. doi:10.1093/rheumatology/key337).

It will be interesting if the authors can show how synovial fibroblasts and macrophages interacts and regulate AXL, sAXL expression. Whether IL-6 signaling is involved in such process, for example by inducing ADAM10/17. The authors showed in their discussion that the role of each cell is not studied. I agree with the authors that identifying the role of each cell (fibroblast or macrophages) from biopsies is challenging. The author can study this point using cell line and/or co-culture of macrophages and synovial fibroblasts. In my opinion, the manuscript needs a more detailed mechanism of how Axl/Mertk modulates the inflammation and IL-6 blockage response.

The title of the manuscript needs to be modified; it implies that AXL/Mertk expression causes the change of the response. Without genetically modified cells or animals it is overreaching to claim that AXL/mertk causes this modulation. Axl/MERTK expression can be a correlation instead of a causation.

The writing of the manuscript at some point is difficult to understand. I would recommend the authors to use simpler sentences (some sentences are 3-4 lines). The words "not surprisingly" can be removed or replaced by "expectedly".

Reviewer #2 (expert in spatial transcriptomics using digital spatial profiling):

This study aims to profile tissue bioresource samples from Rheumatoid arthritis patients to identify novel insights into disease staging/treatment and clinical findings.

Specific comments.

1. abstract - don't abbreviate from the first word (TAM receptors) - define what these are first. This is expanded in the introduction but reading the abstract alone doesn't give the reader sufficient background.

2. Is this a retrospective cohort?

3. Methods - are all arms of this study ethics approved? Or is it only the post-treatment biopsy arm? Unclear from section 4.

4. Were FFPE tissues reviewed by a pathologist? was this double blinded?

5. GeoMx DSP should be Nanostring GeoMx DSP

6. Is Q3 normalisation the recommended way to analyse GeoMx RNA data? there appear to be a number of packages including StandR which are better for data qc/standardisation/normalisation and DEG. - The authors should review this against recent publications as this could significantly bias the results. Have the authors plotted the PCAs to see the groupings?

Figure 8 . The string analysis is useful but hard to visualise - an alternative presentation should be used

Reviewer #3 (expert in TAM receptor kinases):

This study by Verviani expands our understanding of the role of TAM receptors, AXL and MERTK, in the context of rheumatoid arthritis. The authors have presented novel insights into patterns and variation in expression of AXL and MERTK, and their gene partners in samples from rheumatoid arthritis patients to demonstrate clinical relevance of AXL and MER expression based on disease stage and treatment exposure.

Several points would strengthen the manuscript:

- (1) In Figure 1, unique expression patterns are identified in expression of MerTK/Axl and monocyte/macrophage growth factors. Specifically, Axl expression positively correlated with CSF1 gene expression, and MERTK expression positively correlated with CSF1 and CSF2 gene expression. The correlation would be strengthened if protein expression verified these associations.
- (2) In section 2.2, it is stated that the ligands GAS6 and PROS1 genes were in both the Axl and MerTK modules. However, it should be clarified that in the text that GAS6 is a ligand for both MERTK and AXL and that PROS1 is a ligand only for MERTK.
- (3) Figure 3 demonstrates soluble AXL and GAS6 in the synovial fluid of RA patients. As this manuscript is focused on both AXL and MERTK, it would be helpful to determine if sMERTK levels are also increased in patients with synovial inflammation. In a similar manner to AXL, the ectodomain of MERTK is also shed and is a potential ligand sink for Gas6. Were levels of PROS1 in the synovial fluid evaluated?
- (4) Figure 5E evaluates AXL and MERTK protein expression in synovial macrophages following treatment with RTX or TOC. Although it is noted that AXL expression is detected on all synovial macrophages following biological therapies, it should also be clarified that the level of MERTK expression also appeared to be very high in these samples (approx ¾ of these post treatment samples appeared to also express MERTK).
- (5) The data would be enhanced with the addition of single cell RNA sequencing to more clearly define the gene expression patterns in specific cell types.

Minor:

- Supplementary Figure 2 is missing the Figure label 'C'

RESPONSE TO REVIEWERS' COMMENTS

We would like to thank the Reviewers for their valuable time and comments on our manuscript originally submitted with the title "Axl and MerTK modulate synovial inflammation and therapeutic response to IL-6 blockade in rheumatoid arthritis". In order to better take into account the suggestions of the Reviewers, as well as the results of the new experiments carried out post first revision, the title has been slightly modified as follows: "Axl and MerTK regulate synovial inflammation and are modulated by IL-6 inhibition therapy in rheumatoid arthritis".

The Reviewers' comments have been addressed below in a point-by-point response and corresponding changes were made to the revised manuscript as appropriate, highlighted in the track changed version of the manuscript and summarised below. Please, note that the number of pages and lines refer to the tracked-changes version of the manuscript.

Reviewer #1 (expert in interleukin-6, immunoregulation and autoimmunity): The authors using R4RA biopsies shows a potential mechanism of how TAM receptors- AXL and MERTK- can modulate the joint inflammation and response to IL-6 blockade. However, the concept of the synovial AXL expression in modulating the inflammation in mouse and human joints has been addressed including the role of synovial fibroblasts and macrophages (Claire E. et al, *Rheumatology* 2019;58:536-546. doi:10.1093/rheumatology/key337).

We thank the Reviewer for his/her time and constructive comments.

We agree that the work of Claire E. Waterborg and colleagues, published in *Rheumatology* in 2019 has made important contribution to enhancing the understanding of the role of Axl. However, it is uncertain whether their finding in mice showing that Axl is protective for the development of arthritis in the ankle joint compared to the knee is transferable to the human disease, as more than 90% of people with RA acquire foot and ankle symptoms over the course of the disease and in about 20% of patients, foot and ankle symptoms are the first signs of RA.

In addition, while in our manuscript we cite their report that in vitro TGF- β 1 stimulation upregulates Axl in human synovial tissue micromasses containing FLS and macrophages, the enhancing effect of TGF- β 1 on Axl is well known, and we reported consistent findings in co-culture experiments using primary FLS and different subsets of macrophages (MerTK^{neg}CD206^{neg} and MerTK^{pos}CD206^{pos}) in our previous collaborative work with Alivernini S. et al, *Nat Med* 26, 1295–1306 (2020).

The important and novel contribution of our current manuscript is the clinical contextualisation in RA patients' synovial tissue showing indeed a direct correlation between Axl and TGF- β 1, which is maintained / modulated across different stages of the disease i.e., early prior to any treatment exposure, and in established disease following therapeutic intervention. In addition, while Waterborg's work analysed the effect of TGF- β 1 on Axl by cell-cell contact co-cultures, in our manuscript we elucidated the interaction between stromal and myeloid cells and their mutual effects on driving soluble Axl and MerTK shedding in a conditioning media-driven co-culture system, investigating not only the effect of exogenous TGF- β , but also cytokines and growth factors relevant to the inflammatory response in RA, as further discussed in the next section below, and illustrated in the revised results (Figure 4E and 4F, and Supplementary Figure 4) and in the Discussion [page 13, lines 21-27 and page 14, lines -16].

It will be interesting if the authors can show how synovial fibroblasts and macrophages interacts and regulate AXL, sAXL expression. Whether IL-6 signaling is involved in such process, for example by inducing ADAM10/17

The authors showed in their discussion that the role of each cell is not studied. I agree with the authors that identifying the role of each cell (fibroblast or macrophages) from biopsies is challenging. The author can study this point using cell line and/or co-culture of macrophages and synovial fibroblasts. In my opinion, the manuscript needs a more detailed mechanism of how Axl/Mertk modulates the inflammation and IL-6 blockage response.

We thank the Reviewer for this comment. In addition to our related reply above, here we would like to emphasise that, following the advice of the Reviewer about dissecting the interaction between macrophage cell line and synovial fibroblasts, we investigated the effects of RA-derived synovial fibroblasts and macrophages in a conditioning medium-based in vitro system. We demonstrated that LPS-activated RA-derived FLS medium used as conditioning regimen on macrophages, induced a significant and consistent reduction of sMerTK release, while no differences were detected in sAxl and Gas6. Previous collaborative work showed that macrophage membrane MerTK restrains pro-inflammatory effects of FLS, when co-cultured (Alivernini S. et al, Nat Med 26, 1295–1306 (2020)). Here, we added the concept that TLR-activated FLS can thereby affect the release of sMerTK by macrophages, favouring the hypothesis that MerTK (transmembrane)/ligand axis enhancement may counteract unrestrained inflammation.

Notably, we demonstrate an inverse correlation of synovial transcript levels of IL-6 and Axl (Figure 4G) and, importantly, we confirmed the reversal increase of Axl following anti IL-6 therapeutic inhibition with Tocilizumab in vivo in a new analysis using synovial tissue RNAseq data from an independent randomised clinical trial cohort emerging from the STRAP trial, replicating the data observed in the R4RA trial. The clinical results of the STRAP trial have been recently published in the Lancet Rheumatology (Rivellese F, Nerviani A, et al, “Stratification of biological therapies by pathobiology in biologic-naive patients with rheumatoid arthritis (STRAP and STRAP-EU): two parallel, open-label, biopsy-driven, randomised trials, Lancet Rheumatol 2023 5: e648–59). In STRAP, patients failing first line treatment with conventional synthetic DMARDs were randomized to receive either Etanercept, Rituximab, or Tocilizumab. We provide the data below confidentially, as the whole RNAseq data set is not publicly available yet and still subjected to embargo, to further confirm (**Figure 1**, below) that AXL expression is up-regulated in pauci-immune patients, and negatively correlates with IL-6. Moreover, Axl is significantly up-regulated post-tocilizumab (but not post rituximab), as described in the current manuscript.

Figure 1 *Axl* normalised gene expression levels (synovial tissue) in an independent cohorts of RA patients (active)

We have also further analysed the R4RA molecular data to clarify the link between IL-6 signalling and AXL regulation. Interestingly, we found that IL6/AXL ratio is significantly influenced upon treatment with the IL-6-inhibitor tocilizumab, but not affected by the CD20-inhibitor rituximab. This new data has been added to the revised manuscript [Figure 5.E and page 10 lines 18-20]

The title of the manuscript needs to be modified; it implies that AXL/Mertk expression causes the change of the response. Without genetically modified cells or animals it is overreaching to claim that AXL/mertk causes this modulation. Axl/MERTK expression can be a correlation instead of a causation. We would like to thank the Reviewer of his/her comment. We agree and, accordingly, we have modified the title as follows: “Axl and MerTK regulate synovial inflammation and are modulated by IL-6 inhibition therapy in rheumatoid arthritis”,

The writing of the manuscript at some point is difficult to understand. I would recommend the authors to use simpler sentences (some sentences are 3-4 lines). The words "not surprisingly" can be removed or replaced by "expectedly".

We would like to thank the Reviewer of his/her comment. We agree and, accordingly, we have carefully reviewed the manuscript to improve the readability including replacing "not surprisingly" with "expectedly". All changes can be seen in the tracked version of the manuscript.

Reviewer #2 (expert in spatial transcriptomics using digital spatial profiling):

This study aims to profile tissue bioresource samples from Rheumatoid arthritis patients to identify novel insights into disease staging/treatment and clinical findings. We would like to thank the Reviewer for their time and constructive feedback.

Specific comments.

1. abstract - don't abbreviate from the first word (TAM receptors) - define what these are first. This is expanded in the introduction but reading the abstract alone doesn't give the reader sufficient background.

We would like to thank the Reviewer of his/her comment. We agree and we have now modified the abstract accordingly to give the reader a better background about the TAM receptor family [Page 2, line 3]. Unfortunately, as we are allowed only 150 words, we could not provide more accurate information with this word count constraint.

2. Is this a retrospective cohort?

In this manuscript, data from two different cohorts have been presented.

i) The “Pathobiology of Early Arthritis Cohort” (PEAC, <http://www.peac-mrc.mds.qmul.ac.uk>) is an observational study, which started in 2008 and is still open to recruitment. In this work, we analysed data from the first sequential 87 Rheumatoid Arthritis (RA) patients, who had been RNA-sequenced. This cohort is well-established and described in several publications (Lewis, M. J. et al., Cell Rep 28, 2455-2470.e5, 2019; Humby, F. et al., Ann Rheum Dis 78, 761–772, 2019).

ii) The “Rituximab versus tocilizumab in anti-TNF inadequate responder (ir) patients with rheumatoid arthritis” (R4RA, <https://r4ra-nihr.whri.qmul.ac.uk/>) was the first multi-centre synovial biopsy-driven randomised clinical trial recruiting RA patients who had failed the first-line biologic treatment with Tumor Necrosis Factor (TNF) inhibitors; patients underwent a baseline US-guided or arthroscopic synovial biopsy before being randomised to receive either rituximab (RTX) or intravenous tocilizumab (TOC). The study ran between 2013 and 2019; the clinical results of the trial were published in 2021 (Humby, F. et al., The Lancet 397, 305–317, 2021), while the molecular analysis was published in 2022 (Rivellese, F. et al., Nat Med 28, 1256–1268, 2022). In this manuscript, we focussed on the expression and interpretation of the role of the TAM receptor family in established RA.

Both cohorts have been described in the “Methods” section. We have made some addition [Page 18 between line 4 and 34, and page 19 between line 1 and 3] to better clarify the features of the cohorts based on your comment above as well as including clearer data on the patients’ sex.

3. Methods - are all arms of this study ethics approved? Or is it only the post-treatment biopsy arm? Unclear from section

As per good clinical practice, both the studies included in this manuscript (see above) received ethics approval to all arms. In both studies, all patients signed an informed consent before being recruited and undergoing the baseline synovial biopsy (mandatory) and/or the optional post-treatment biopsy. Notably, as approved by the ethics committee, all patients with their consent agreed to allow additional research related to disease pathobiology to be carried out on the donated samples. The REC numbers have been reported in Methods [Page 18, lines 20-21 and page 19, lines 1-2]

To avoid confusion, we have moved the sentence “An optional post-treatment biopsy of the same joint is repeated 6-months after starting the treatment” up [Page 18, now lines 31-32] so that it is not immediately followed by the sentence “The study was approved by the National Research Ethics Service Committee London Dulwich (REC 05/Q0703/198)”.

For added clarity, following the Reviewer’s question, we modified this sentence as below [Page 18, lines 33-34]:

“The study protocol, *including baseline and optional repeated synovial biopsy*, was approved by the institutional review board of each study centre or relevant independent ethics committees (UK Medical Research and Ethics Committee (MREC) reference: 12/WA/0307).”

4. Were FFPE tissues reviewed by a pathologist? was this double blinded?

PEAC semiquantitative immuno-scores and pathotype classification were performed by two researchers expert in synovial pathology at Queen Mary University of London (QMUL). R4RA patients were initially scored in the pathology laboratory of Barts Health NHS Trust (London, UK) by a consultant pathologist; histological scores were replicated at QMUL by an independent expert. In both cases, discrepancies in classification were resolved through mutual agreement. The sentences above have been added to Methods [Page 19, lines 16-21].

5. GeoMx DSP should be Nanostring GeoMx DSP

This has been amended throughout the manuscript.

6. Is Q3 normalisation the recommended way to analyse GeoMx RNA data? there appear to be a number of packages including StandR which are better for data qc/standardisation/normalisation and DEG. - The authors should review this against recent publications as this could significantly bias the results. Have the authors plotted the PCAs to see the groupings?

The Nanostring GeoMx RNA data normalization was performed by our collaborators at Nanostring according to their recommended and validated normalization method used in trade-mark commercial products. Supplementary Figure 7 in the manuscript now illustrates the PCA of the Q3 normalized data where the samples are shown to be grouped by tissue region. Furthermore, the groupings by patients who responded to treatment or are refractory were illustrated for each tissue region of interest (ROI).

Following the Reviewer's suggestion, the Nanostring GeoMx RNA data was also explored using the StandR package (v1.4.2) on R version 4.3.1, which was installed on 27 June 2023. Since then, the manuscript accompanying standR has been published only recently on Nucleic Acids Research on 11th November 2023 and the package has been continuously revised on Github over the past six months with the latest version on Bioconductor being v1.6.0. This suggests that the package is still being updated and perfected and we found that the package is not completely stable.

Notably, our analysis using StandR package (v1.4.2) demonstrated no significant deviations from our normalized GeoMx RNA data provided by Nanostring. We note that previous versions of StandR on older R versions before 4.3.1 (especially StandR v.1.0.0 on R version 4.2.0) are not viable due to various glitches that have been recently fixed with the latest standR version.

Following the QC functions in the standR package, we found the following (**Figure 2 below**):

- No genes have an expression value below the default threshold (min count=5) in more than the default percentage of regions of interest (90%) using the addperROIQC function. The plotGeneQC function illustrates that the small percentage of lowly-expressed genes in each sample.
- The distribution of nuclei count and library size is relatively smooth and no samples should be filtered out due to low library size (defined by 50,000 in their quick guide) using the plotROIQC function.

Gene QC

ROI level QC

Figure 2 Quality check (QC) of GeoMx digital spatial profiling (DSP) data by StandR

The top plot visualises the metadata where 72 regions of interest (ROI) (aggregate, lining and sublining) were profiled in synovial biopsies from rheumatoid arthritis patients (4 refractory and 8 responders to rituximab or tocilizumab). The plot for gene QC illustrates the percentage of lowly-expressed genes in each sample. The plot for ROI level QC illustrates the distribution of nuclei count and library size where a low library size is defined by a cut-off 50,000.

Various normalization methods are offered by StandR. We have plotted the PCA using the GeoMX RNA data normalized by standR (CPM, TMM, size factors and upper quartile) (**Figure 3 below**) showed similar PCA distributions to our normalized GeoMx RNA data. It is important to note that the normalized data originally reported, as mentioned above, was provided directly by our Nanostring collaborators using their standard pipeline to normalize the data.

Figure 3 Clustering of the regions of interest in synovial biopsies from rheumatoid arthritis patients using various normalization methods by StandR. Principal component analysis (PCA) of the digital spatial profiling (DSP) of 72 regions of interest (aggregate, lining and sublining) in synovial biopsies from rheumatoid arthritis patients where the data has been normalized counts per million (CPM), trimmed mean (TMM), size factors and upper quartile by StandR.

Our conclusion from using standR is that other normalisation methods e.g. TMM, sizeFactors give very similar results that do not change the conclusions of the analysis. Genes of interest between responders and refractory patients for each tissue region of interest using the TMM normalized data by standR generated similar boxplots and statistical significance compared to the boxplots shown in the manuscript using our normalized GeoMx RNA data provided by Nanostring (**Figure 4 below**).

Q3 normalisation (Manuscript figure)

Figure 4 Assessing genes of interest between responders and refractory rheumatoid arthritis patients for each region of interest. Boxplots of selected genes of interest in aggregate, lining and sublining areas of responders and refractory patients using GeoMx RNA data normalized by trimmed means (TMM) using StandR or by Q3 provided by Nanostring. Wilcoxon test was undertaken in comparing responders and refractory patients.

Figure 8 . The string analysis is useful but hard to visualise - an alternative presentation should be used This has been corrected in the revised version of the manuscript, where we have enlarged and highlighted the “center” of the string (Axl or MerTK).

Reviewer #3 (expert in TAM receptor kinases):

This study by Verviani expands our understanding of the role of TAM receptors, AXL and MERTK, in the context of rheumatoid arthritis. The authors have presented novel insights into patterns and variation in expression of AXL and MERTK, and their gene partners in samples from rheumatoid arthritis patients to demonstrate clinical relevance of AXL and MER expression based on disease stage and treatment exposure.

We thank the reviewer for their time, comments, and suggestions.

Several points would strengthen the manuscript:

(1) In Figure 1, unique expression patterns are identified in expression of MerTK/Axl and monocyte/macrophage growth factors. Specifically, Axl expression positively correlated with CSF1 gene expression, and MERTK expression positively correlated with CSF1 and CSF2 gene expression. The correlation would be strengthened if protein expression verified these associations.

We agree with the Reviewer that the analysis at protein level would further strengthen our observations. However, in the absence of this data, mostly due to the limitation of quantifying soluble mediators in FFPE tissue, we took a different approach and to, corroborated the results, we analysed an independent RNAseq dataset (replication cohort) from synovial biopsies of patients recruited into the “Stratification of biological therapies by pathobiology in biologic-naïve patients with rheumatoid arthritis (STRAP and STRAP-EU)” (Rivellese F, Nerviani A, et al. Lancet Rheumatol 2023 5: e648–59). This third different population cohort includes patients who failed first line treatment with conventional synthetic DMARDs, but biologic naïve, who were randomized to receive either Etanercept, Rituximab, or Tocilizumab. As shown, we could confirm that the same correlations observed in PEAC and R4RA were also maintained in this independent cohort. As the RNAseq dataset is not publicly available yet and still subjected to embargo, we provide this data confidentially.

Figure 5 AXL, MERTK, CSF1, CSF1R, CSF2, and CSF2RA synovial expression in an independent cohort of RA patients (active)

(2) In section 2.2, it is stated that the ligands GAS6 and PROS1 genes were in both the Axl and MerTK modules. However, it should be clarified that in the text that GAS6 is a ligand for both MERTK and AXL and that PROS1 is a ligand only for MERTK.

We would like to thank the reviewer for this comment. We have clarified this concept “Axl/MerTK ligand GAS6 and MerTK ligand Protein S (PROS1)” in section 2.2 [Page 6, line 10].

(3) Figure 3 demonstrates soluble AXL and GAS6 in the synovial fluid of RA patients. As this manuscript is focused on both AXL and MERTK, it would be helpful to determine if sMERTK levels are also increased in patients with synovial inflammation. In a similar manner to AXL, the ectodomain of MERTK is also shed and is a potential ligand sink for Gas6. Were levels of PROS1 in the synovial fluid evaluated?

We would like to thank the reviewer for the suggestion. Due to limited availability of the synovial fluid, we prioritized the quantification of soluble MerTK. For consistency, we also repeated the quantification of sAxl and Gas6. New data are presented in Figure 3F, 3G, and 3H and in Supplementary Figure 3. Briefly, sMerTK was not significantly increased in patients with a higher degree of inflammation, although there was a trend. We still observed a trend of significant positive correlation between sAxl and Gas6, but this was not statistically significant. The synovial fluid data are now complemented by the in vitro data showing how sMERTK release by macrophages can be modulated by TLR-activated synovial fibroblasts.

(4) Figure 5E evaluates AXL and MERTK protein expression in synovial macrophages following treatment with RTX or TOC. Although it is noted that AXL expression is detected on all synovial macrophages following biological therapies, it should also be clarified that the level of MERTK expression also appeared to be very high in these samples (approx $\frac{3}{4}$ of these post treatment samples appeared to also express MERTK).

We would like to thank the reviewer for this comment. We agree that the presence of MerTK in about 75% of macrophages before and after treatment should be noted. We have therefore added the following sentence “Notably, about 75% of macrophages pre- and post-treatment were MerTK+ (Figure 5.F)” [page 10, line 25-26]

(5) The data would be enhanced with the addition of single cell RNA sequencing to more clearly define the gene expression patterns in specific cell types. Thank you for your comment. Certainly the addition of single cell RNA sequencing would further enhance the overall understanding of the gene expression patterns. Previous works have already, at least partially, assessed this aspect (e.g., Alivernini S. et al., Nat Med 26, 1295–1306. 2020; Zhang, F. et al., Nat Immunol 20, 928–942 2019). We have indeed reported some analysis extrapolated from single cell data published by Zhang (Supplementary Figures 2C and 6B).

In our manuscript, we focused on large cohorts of whole synovial tissue RNAseq, which could be integrated with high-quality clinical and histological data. Although single cell data are crucial to understand the heterogeneity of various cell subsets, it cannot be a substitute for the information provided by the whole tissue both in terms overall molecular pathology and spatial identity definition. As we did not have tissue available to generate single-cell RNAseq data, we set up in vitro systems as an alternative approach to study the effect of various cell types (fibroblasts and macrophages) on TAM receptors expression, as discussed in our reply to Reviewer 1. New data have been reported in Figure 4E and 4F, and Supplementary Figure 4.

Minor:

- Supplementary Figure 2 is missing the Figure label ‘C’

Thank you for noticing it. We have now added label C to the Figure.

REVIEWERS' COMMENTS

Reviewer #1 (Remarks to the Author):

The authors have addressed my comments properly. I have no further comments.

Reviewer #2 (Remarks to the Author):

The authors have addressed my concerns

Reviewer #3 (Remarks to the Author):

The authors have adequately addressed the key points of my comments, resulting in an improved manuscript. I believe this manuscript is acceptable for publication.

I understand that technical limitations prevented the authors from perform certain experiments, such as investigating protein expression levels in FFPE tissue. Accordingly, I appreciate the effort to confirm the observed gene expression correlations in an independent dataset, which further strengthens the author's claims. Given the limited availability of synovial fluid, the quantification of sMERTK levels adequately addresses our comment (3). Additionally, I appreciate that the authors addition of new in vitro data that provide further insight into the roles of AXL and MERTK in macrophages and fibroblasts, as well as the differential regulation of their shedding from macrophages.

Minor point: in section 2.4 line23, the authors state "there were no differences in sAxl/Gas6 or sMerTK/Gas6 ratio 24 (Supplementary Figure 4.B)." However, I could not find this comparison in Sup Fig 4B. Please clarify.

RESPONSE TO REVIEWERS' COMMENTS

Reviewer #1 (Remarks to the Author):

The authors have addressed my comments properly. I have no further comments.

We thank the Reviewer for his/her comment.

Reviewer #2 (Remarks to the Author):

The authors have addressed my concerns

We thank the Reviewer for his/her comment.

Reviewer #3 (Remarks to the Author):

The authors have adequately addressed the key points of my comments, resulting in an improved manuscript. I believe this manuscript is acceptable for publication.

I understand that technical limitations prevented the authors from perform certain experiments, such as investigating protein expression levels in FFPE tissue. Accordingly, I appreciate the effort to confirm the observed gene expression correlations in an independent dataset, which further strengthens the author's claims. Given the limited availability of synovial fluid, the quantification of sMERTK levels adequately addresses our comment (3). Additionally, I appreciate that the authors addition of new in vitro data that provide further insight into the roles of AXL and MERTK in macrophages and fibroblasts, as well as the differential regulation of their shedding from macrophages.

Minor point: in section 2.4 line23, the authors state "there were no differences in sAxl/Gas6 or sMerTK/Gas6 ratio 24 (Supplementary Figure 4.B)." However, I could not find this comparison in Sup Fig 4B. Please clarify.

We thank the Reviewer for their comment. The comparison was not shown indeed, thank you for noticing it. The sentence has been removed accordingly.